# Concept Bottleneck Generative Models

**Aya Abdelsalam Ismail**[1,2] *   **Julius Adebayo**[3] *   **Héctor Corrada Bravo**[1]
**Stephen Ra**[1,2]   **Kyunghyun Cho**[1,2,4,5]
[1]Genentech   [2]Prescient Design
[3]Guide Labs
[4]Department of Computer Science, New York University
[5]Center for Data Science, New York University

## ABSTRACT

We introduce a generative model with an intrinsically interpretable layer—a concept bottleneck layer[†]—that constrains the model to encode human-understandable concepts. The concept bottleneck layer partitions the generative model into three parts: the pre-concept bottleneck portion, the CB layer, and the post-concept bottleneck portion. To train CB generative models, we complement the traditional task-based loss function for training generative models with a concept loss and an orthogonality loss. The CB layer and these loss terms are model agnostic, which we demonstrate by applying the CB layer to three different families of generative models: generative adversarial networks, variational autoencoders, and diffusion models. On multiple datasets across different types of generative models, steering a generative model, with the CB layer, outperforms all baselines—in some cases, it is *10 times* more effective. In addition, we show how the CB layer can be used to interpret the output of the generative model and debug the model during or post training.

## 1 INTRODUCTION

Improvements in generative modeling have led to these models being applied to produce photo realistic images (Saharia et al., 2022), video (Ho et al., 2022; Villegas et al., 2022), protein sequences (Ingraham et al., 2022), small molecules (De Cao and Kipf, 2018), and coherent text (Brown et al., 2020). However, current generative models admit little-to-no-room for interpretation, which limits the ability to fix these models when they make mistakes. Consider a model trained to generate protein sequences; a domain expert might be interested in determining whether the model has captured desirable features like thermostability and toxicity. An important goal is to use these features, as knobs, to steer the model to generate sequences that satisfy desired ranges of thermostability and toxicity. In this work, we develop generative models with *intrinsically interpretable components* that can be used to simultaneously *interpret, debug, and steer* the output of the model.

**Challenges with interpreting and steering generative model representations.** Current approaches for interpreting generative models cannot reliably indicate that a model's representations map to human-understandable features that the model relies on for its output. One approach (Belinkov, 2022) for interpreting a generative model's representations uses a low-complexity model to predict a human understandable feature from the model's representations. However, high predictive performance of the low-complexity model does not indicate that the model's output relies on that feature (Lovering and Pavlick, 2022). Another approach constrains the model to learn *disentangled* (Higgins et al., 2017; Tran et al., 2017; Meo et al., 2023) representations—that is, representations that can be decomposed into independent factors. Nevertheless, it is not possible to guarantee that a disentangled representation is human interpretable (Locatello et al., 2019). Other approaches project model representations into lower dimensions and then search for directions correlated with human interpretable features (Härkönen et al., 2020); yet since the model was not constrained to learn such features, it is possible that its representations do not encode these desired features.

---

* The authors have contributed equally to this work and are listed in alphabetical order.
† Code is available at https://github.com/prescient-design/CBGM

**(a) CB-StyleGAN2 on FFHQ**

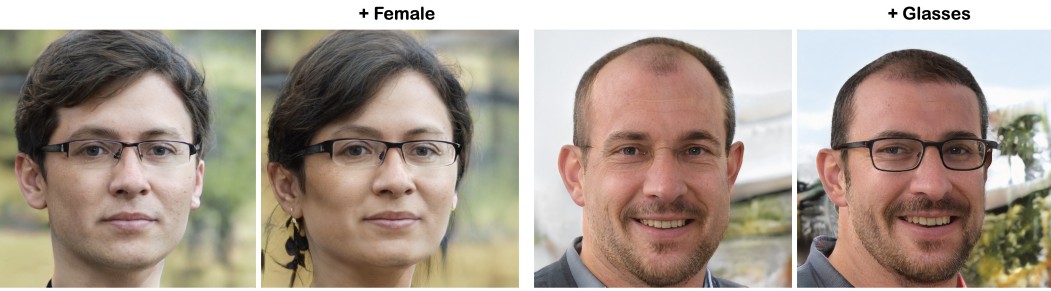

**(b) CB-DDPM on LAION Aesthetics**

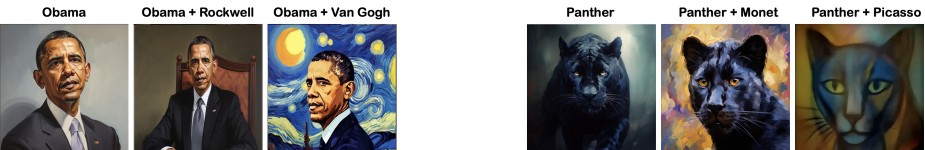

Figure 1: Steering using the CB layer for different architectures and datasets. The CB layer does not hurt generation quality and can scale to high-dimensional settings. (a) A concept bottleneck StyleGAN2 model trained on the FFHQ 1024x1024 dataset. (b) A concept bottleneck diffusion model trained on the LAION Aesthetics 256x256 dataset, where we steer the artistic style.

**Concept bottleneck generative models (CBGMs).** To address challenges with interpreting and steering current generative models, we present concept bottleneck generative models. In this generative model, one of its internal layers—a concept bottleneck (CB) layer—is constrained to encode human-understandable features. We insert the CB layer into the generative model's architecture, the generative model can be divided into three parts: the pre-concept bottleneck portion, the CB layer, and the post-concept bottleneck portion. The pre-concept bottleneck portion maps from the input to activations, which the CB layer then maps into human-understandable features. The pre-defined concepts alone can be incomplete, so we allow additional representational capacity for unknown concepts that are constrained to be orthogonal to the pre-defined features. Lastly, the post-concept bottleneck layer maps both the output of the CB layer and unknown concepts to the generated output. To train CBGMs, we complement the traditional loss functions for training generative models with two terms: a concept loss and an orthogonality loss. The CB layer is model agnostic, which we apply to 3 different families of generative models: generative adversarial networks, variational autoencoders, and diffusion models. Using the CB layer, we can demonstrate the following capabilities:

- **Steering Generative Models:** By intervening on the output of the CB layer, we can modulate the level of a particular concept present in the output of a generative model. We use this capability to control single concepts independently, and multiple concepts simultaneously. Figure 1 shows an example of intervening on denoising diffusion probabilistic models and generative adversarial networks with a CB layer. We can change the concept of the image while preserving the image quality. Across several datasets and 3 types of generative models, steering a generative model with the CB layer outperforms several input conditioning baselines. We find that when the number of concepts increases the performance of conditioning models dramatically degrades while the performance of CBGMs remains almost unchanged; in such cases, we can achieve up to 10x improvement in steerability compared to traditional approaches.

- **Understanding and Debugging Generative Models:** The CB layer can also be used to debug a generative model during and post training. In Section 4.3, we show that the CB layer can help identify the important concepts that are responsible for a model's generated output. Similarly, the output of the concept bottleneck (ICB) layer can be used to distinguish a model that has learned the pre-defined human-understandable features from a model that has not.

## 2 SETUP & BACKGROUND

We now give an overview of concept bottleneck models and the types of generative models that we consider. We assume that all training samples come with pre-defined human understandable features as such: $\{(x_i, c_i)\}_{i=1}^n$, where an example $x_i \in \mathbb{R}^d$ and an associated concept $c_i \in \mathbb{R}^k$ with $k << d$. Generative models consider the task of modeling the probability distribution of an observed random variable $\mathbf{x} \sim \mathbb{P}(\mathbf{x})$. We consider three generative model families: variational autoencoders (VAEs) (Kingma and Welling, 2013), generative adversarial networks (GANs) (Goodfellow, 2016), and diffusion models (Sohl-Dickstein et al., 2015; Song et al., 2020; Ho et al., 2020).

**Concept Bottleneck Models (CBMs) & Concept Embedding Models (CEMs).** CBMs (Koh et al., 2020) inserts an 'interpretable' layer into a deepnet. Specifically, CBMs map samples $x_i$ to labels $y_i$ by first mapping $x_i$ to an intermediate representation $c_i = h(x_i)$, where $c_i$ are understandable human concepts (e.g., hair color). An 'interpretable' label predictor, $f$, then maps the predicted concepts to labels: $y = f(h(x))$. Consequently, the goal is to learn the following relationship $x_i \xrightarrow{h} c_i \xrightarrow{f} y_i$. The functions $h$ and $f$ can be learned jointly, sequentially, or independently. To improve the expressiveness of CBMs, (Espinosa Zarlenga et al., 2022) proposed CEMs, which map the input to a high-dimensional representation for each concept.

**Variational Autoencoder.** VAEs (Kingma and Welling, 2013; Kingma et al., 2019) consists of two components: an encoder and a decoder. The encoder, $q(z_i|x_i)$, maps an input, $x_i$, to a distribution over the latent variable, $z_i$. The encoder parameterizes a Gaussian density, from which we sample a latent vector $z_i^d$. VAEs regularizes this latent distribution to be similar to the prior distribution $p(z)$, which is typically $z \sim \mathcal{N}(0, I)$. The vector $z_i^d$ is passed to the decoder, $p(x_i|z_i^d)$, a function that maps the latent vector to a distribution over the input. The task loss for the VAE is the negative log-likelihood:

$$\mathcal{L}(x_i)_{\text{task,vae}} = D_{KL}\big(q\left(z_i|x_i\right)||p\left(z_i\right)\big) - \mathbb{E}\big[\log p(x_i|z_i^d)\big], \tag{1}$$

where $D_{KL}$ is the Kullback-Leibler divergence.

**Generative Adversarial Networks.** GANs (Goodfellow et al., 2014) consist of a generator $G$ that captures the data distribution and a discriminator $D$ that estimates the probability that a sample came from either the training data or $G$. The generator maps a noise vector $z_i$ to an output $\hat{x}_i = G(z_i)$. Conditional GANs (Mirza and Osindero, 2014; Chen et al., 2016; Odena et al., 2017) augment the input to the generator with the concepts and also learn $G(z_i|c_i)$. They, however, do not learn $c_i = h(z_i)$. To learn $c_i = h(z_i)$, we first encode $x_i$ into a latent vector $q(z_i|x_i)$, since $p(c_i|x_i)$ is known, i.e., concepts for a given sample are known, we now can learn $c_i = h(z_i)$. Similar to VAEs, we then sample a new latent vector $z_i^d$ and use this for generation $G(z_i^d)$. This approach has been widely employed (Larsen et al., 2016; Isola et al., 2017; Wang et al., 2018) for image generation. For training, we use the loss introduced by VAE-GANs (Larsen et al., 2016), which combines the VAE encoder regularization prior loss with a GAN loss.

$$\mathcal{L}(x_i)_{\text{task,gan}} = D_{KL}\big(q\left(z_i|x_i\right)||p\left(z_i\right)\big) + \mathbb{E}\big[\log D\left(x_i\right)\big] + \mathbb{E}\big[1 - \log D\left(G\left(z_i^d\right)\right)\big]. \tag{2}$$

**Diffusion models.** Diffusion models can be interpreted as latent variable models with two stages (Sohl-Dickstein et al., 2015; Ho et al., 2020). Given input data $x_i$, the first stage is the forward diffusion process, which involves incrementally adding Gaussian noise to the input and can be described as: $q\left(x_i^t|x_i^{t-1}\right) = \mathcal{N}\left(x_i^t; \sqrt{1 - \beta_t}x_i^t, \beta_t I\right)$, where $\beta_t$ is determined according to a pre-specified schedule. The second stage of the process learns a denoising model, $p\left(x_i^{t-1}|x_i^t\right)$ that reverses the forward process. The model is trained to maximize a lower bound to the marginal likelihood, which can be relaxed into a mean-squared error loss as:

$$\mathcal{L}(x_i^t)_{\text{task,df}} = \sum_{i=1}^T \mathbb{E}\big\|\mu(x_i^t, t) - \hat{\mu}\left(x_i^t, x_i^{t=0}\right)\big\|^2. \tag{3}$$

We refer to (Ho et al., 2020; Weng, 2021; Rogge and Rasul, 2022) for a more detailed overview of diffusion models.

## 3    CONCEPT BOTTLENECK GENERATIVE MODELS

We propose to insert a concept embedding layer—which we term a concept bottleneck (CB) layer—into a generative model. Our overall framework, shown in Figure 2, consists of 3 parts: the portion of the generative model before the CB layer (the pre-concept bottleneck network); the CB layer, and the portion of the generative model after the CB layer (the post-concept bottleneck network). The pre-concept bottleneck and post-concept bottleneck networks are specific to the type of model (GANs, diffusion, & VAE) used for generation, while the CB layer is common across all generative model families. We now discuss the architecture of the CB layer, its loss functions, and an intervention procedure for steering the generative model.

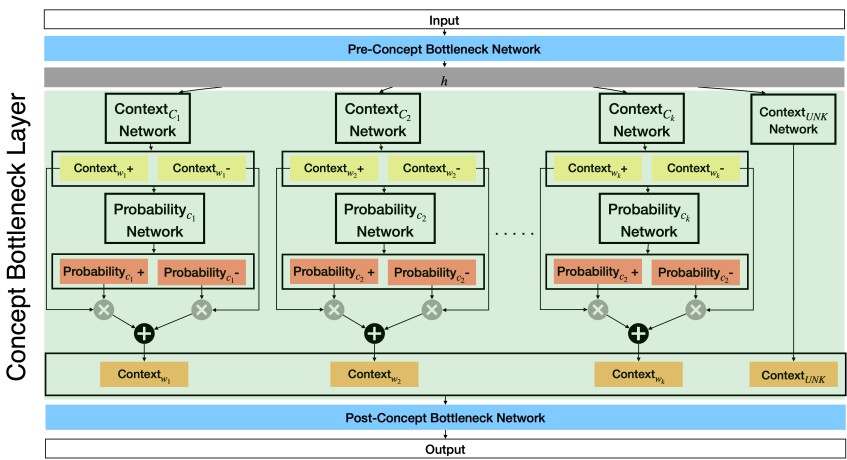

Figure 2: Schematic of the concept bottleneck layer for a generic generative model.

### 3.1    ARCHITECTURE

We adopt the CEM layer of Espinosa Zarlenga et al. (2022) to the generative model setting. However, unlike in supervised learning tasks where the concept set is assumed to be near complete (Koh et al., 2020; Espinosa Zarlenga et al., 2022; Yuksekgonul et al., 2022), it is unrealistic to expect the pre-defined human understandable features—concepts—to be complete in the generative setting. For example, consider generating a human face; finding a comprehensive set of concepts that can control every generation aspect is challenging. However, one might have available concepts such as hair color, eye color, and skin tone. We extend the CEM layer of Espinosa Zarlenga et al. (2022) to encode for unknown concepts that might also be required for generation. The proposed modification of the concept bottleneck layer is shown in Figure 2, and consists of $k$ concept networks and an extra unknown-concept network.

**Concept Embeddings.**    Each concept is represented with two embeddings: $w_i^+, w_i^- \in \mathbb{R}^m$ representing the active and inactive concept states, respectively. The output of the pre-concept bottleneck portion, an embedding vector $h$, is fed into a context network, $\phi$, that maps $h$ into two embeddings per concept. This context network can be viewed as two separate functions: $w_i^+ = \phi^+(h)$ and $w_i^- = \phi^-(h)$. Embeddings $w_i^+$ and $w_i^-$ are encouraged to be aligned with ground-truth concept $c_i$ by the function $\Psi_i$ trained to predict the probability of concept $c_i$ being active from the joint embedding space, $\hat{c}_i = \Psi_i([w_i^+, w_i^-]^T) \in [0, 1]$ (i.e, $\hat{c}_i$ is the predicted probability of concept $c_i$). The final context embedding, $w_i$, is a weighted mixture of the two embedding $w_i = (\hat{c}_i w_i^+ + (1 - \hat{c}_i) w_i^-)$. All $k$ concept embeddings are concatenated together, along with the non-concept embeddings, $w = [w_1, w_2, \ldots, w_{k+1}]$, resulting in a bottleneck $f(h) = w$ with size $m(k + 1)$, which is fed into the post-concept network.

**Adding the CB layer.** We make the CB layer to be the first layer of the decoder for a VAE. Diffusion models typically follow a U-Net structure, and the CB layer is inserted after the middle block of the U-Net. For GANs, we make the input to the generator to be output of the CB layer. We refer to Figure 6 in the Section A of the Appendix for additional discussion.

**Intervention on Concept Probabilities.** CBGMs support test-time interventions, which allows the user to steer the output of the generative model. To intervene on concept $c_i$, one replaces the probability of the concept being active, $\hat{c}_i$, with the desired probability $\bar{c}_i$. The CB layers' context embedding, $w_i = \left( \bar{c}_i w_i^+ + (1 - \bar{c}_i) w_i^- \right)$, is combined as a convex combination of the positive and negative context vectors. The new context embedding will then be passed to the post-concept bottleneck model to generate the new output with the desired concept.

## 3.2 LOSS FUNCTIONS & TRAINING

We train CBGMs in an end-to-end fashion by jointly minimizing the following loss function:

$$\mathcal{L}_{\text{total}} = \mathcal{L}_{\text{task}} + \alpha \mathcal{L}_{\text{con}} + \beta \mathcal{L}_{\text{orth}} \tag{4}$$

where $\mathcal{L}_{\text{task}}$ is the task loss, $\mathcal{L}_{\text{con}}$ is the concept loss, and $\mathcal{L}_{\text{orth}}$ is the concept orthogonality loss. The hyperparameters $\alpha$, and $\beta$ control the relative importance of the concept, and orthogonality losses. While the task loss is specific to each generative model family, $\mathcal{L}_{\text{con}}$, and $\mathcal{L}_{\text{orth}}$ are common across different models.

- **Task Loss**: the task loss is the traditional objective function for each generative model family as defined in Equations 1, 2, and 3, respectively.
- **Concept Loss**: across all generative model classes, the concept loss, $\mathcal{L}_{\text{con}}$, is the binary cross-entropy loss on the output of each probability network of the CB layer.
- **Concept Orthogonality Loss**: An undesirable situation occurs if the unknown concepts are transformations of the known concepts, which hampers the ability to control the model's output. To prevent this, we encourage the unknown concepts to be orthogonal to the outputs of each concept network using an orthogonality constraint (Ranasinghe et al., 2021) that minimizes the cosine similarity between the concept context embedding and the unknown context embedding as follows:

$$\mathcal{L}_{\text{orth}} = \sum_{j \in B} \frac{\sum_{i=1}^{i=k} \left| \langle w_i \,, \, w_{k+1} \rangle \right|}{\sum_{i=1}^{i=k} 1} \tag{5}$$

where $\langle \cdot \,, \, \cdot \rangle$ is the cosine similarity applied to two embedding, $| \cdot |$ is the absolute value, and $B$ denotes mini-batch size and $j$ denotes each sample in the mini-batch. The cosine similarity in the above equation involves the normalization of features such that $\langle x_i \,, \, x_j \rangle = \frac{x_i \cdot x_j}{\|x_i\|_2 \cdot \|x_j\|_2}$, where $\| \cdot \|_2$ is $l_2$ norm.

## 4 EXPERIMENTS & RESULTS

In this section, we answer the following:

1. **Steerability**: How well does the CB layer control the output of a generative model, and how does steering by intervening on the CB layer compare to traditional conditional generation methods?
2. **Interpretability**: How can the CB layer be used to help interpret and debug a generative model?
3. **Generation quality & Ablations** How does the inclusion of the CB layer affect generation quality? Also, how sensitive is the model's steerability to each of the proposed components?

## 4.1 SETUP

**Datasets.** We consider the following datasets: (a) The $64 \times 64$ version of CelebFaces Attributes (Celeb-A), (Liu et al., 2015) that is annotated with 40 attributes (e.g., male, smiling etc.); (b) a curated subset of 'aesthetic' LAION (Schuhmann et al., 2022) dataset that is annotated with concepts corresponding to one of three categories: artistic style, presence of a famous person, and fictional characters giving a total of 155 concepts; (c) The Caltech-UCSD birds species (CUB) (Wah et al., 2011) that comes annotated with 312 concepts; (d) FlickrFaces-HQ (FFHQ) human faces dataset (Karras et al., 2019) consisting of 70,000 high-quality images at $1024^2$ resolution annotated with 8 concepts describing the face (e.g., female, smiling, lipstick etc.); and (e) the Color-MNIST dataset (Deng, 2012), where we take each label category and color as concepts.

**Generative Model Baselines.** We consider three types of generative models: VAE, GANs, and diffusion models. We compare our method with each family's most commonly used conditional generation approach. For GANs, we consider CGAN (Mirza and Osindero, 2014), and ACGAN (Odena et al., 2017). For diffusion models, we compare with classifier (Dhariwal and Nichol, 2021) and classifier-free (CF) guidance (Ho and Salimans, 2022). For VAEs, we benchmark against conditional VAEs (CVAE) (Sohn et al., 2015).

## 4.2 STEERING CB GENERATIVE MODELS

We assess the ability of the CB layer to steer the output of the generative model effectively. As discussed in Section 3, to 'turn on' a concept, we intervene on the concept probability, $c_i$. We compare concept intervention via the CB layer to other prominent conditional generation strategies.

**Experimental Setup** We train concept classifiers to detect the presence of a concept in an input on the real data; we ensure that the minimum accuracy of any concept classifier is at least 98% on a held-out test set from the real data. We sample noise latent vectors and then use them to generate images from the model. For each concept, we pass the generated image to the concept classifier, save the latent vector if the classifier predicts the concept to be absent (i.e., the probability of the concept present is less than 0.5), until we have 1000 samples. For each steering approach, we intervene to 'turn-on' the concept and generate new outputs. Finally, we assess the newly generated images with the original classifier again and measure the fraction of inputs for which it predicts the concept to be present; we termed this metric the steerability accuracy. We examine the performance of models in two regimes: (a) *Small balanced concepts regime:* We extract a subset of concepts with balanced labels across the dataset; for Celeb-A we extract the 8 most balanced concepts, and for CUB, we extract the 10 most balanced concepts. We then train models on the balanced subset. (b) *Large unbalanced concepts regime:* We train on all concepts regardless of their distribution in the dataset; here, we considered all 40 concepts for Celeb-A. We repeat each experiment 3 times and report the mean and variance in Table 1 (per concept metrics is available in Appendix ).

| Concept Regime | Small balanced concepts | | Large unbalanced concepts |
|---|---|---|---|
| Dataset | CUB (10 concepts) | Celeb-A (8 concepts) | Celeb-A (40 concepts) |
| CGAN | $5.4 \pm 0.4$ | $8.7 \pm 1.3$ | $2.9 \pm 0.0$ |
| ACGAN | $18.5 \pm 0.4$ | $9.2 \pm 0.7$ | $1.2 \pm 0.1$ |
| CB-GAN | $\mathbf{21.3 \pm 0.3}$ | $\mathbf{25.6 \pm 0.5}$ | $\mathbf{23.1 \pm 0.2}$ |
| CF-Diffusion | $2.7 \pm 1.9$ | $7.15 \pm 3.8$ | $5.1 \pm 2.4$ |
| CG-Diffusion | $2.1 \pm 1.4$ | $6.8 \pm 1.1$ | $5.4 \pm 2.6$ |
| CB-Diffusion | $\mathbf{14.8 \pm 6.2}$ | $\mathbf{13.8 \pm 2.7}$ | $\mathbf{12.6 \pm 1.7}$ |
| CVAE | $1.2 \pm 0.1$ | $5.9 \pm 1.6$ | $5.3 \pm 1.6$ |
| CB-VAE | $\mathbf{10.7 \pm 4.6}$ | $\mathbf{16.3 \pm 4.1}$ | $\mathbf{15.5 \pm 3.3}$ |

Table 1: Average steerability metric for all concepts across three generative model families.

**Results** In the small balanced concepts regime, we find that steering via the CB layer is much more effective across different datasets and generative models when compared to different standard conditioning approaches, in some cases up to 10x more effective, as shown on the CUB dataset for diffusion and VAEs. In the large unbalanced concepts regime, we find that the performance of conditioning models dramatically decreases with the increase in the number of concepts for example ACGAN accuracy drops from 9.2% to 1.2% when the model is required to learn 40 concepts instead of 8 concepts; however using a CB layer the performance degradation is minimal (at most 2.5%).

**Scaling Input Size & Number of Concepts** We now test the effectiveness of CB generative models as we scale the input size, and the number of concepts. First, insert a CB layer into the StyleGAN2 architecture and train the model on FFHQ $1024^2$ dataset, Figure 1 (a) shows samples generated from the model. We find that adding a concept bottleneck layer does not affect the generation quality, and we are able to steer different facial concepts. Second, to test the effect of scaling the number of concepts, we train a CB diffusion model on a curated subset of the LAION dataset, where each image is annotated with 155 concepts. In Figure 1 (b), we demonstrate the effect of steering the artistic style. The first image in each example shows generated inputs without any concept intervention, while

subsequent columns hold object attributes fixed and intervene in various artistic styles. These results suggest that concept bottleneck generative models can scale to contemporary settings.

### 4.3 INTERPRETABILITY

We now show how the concept-bottleneck layer can be used to (a) interpret the output of a generative model and (b) debug the model during and after training.

#### 4.3.1 INTERPRETING THE OUTPUT OF A CB GENERATIVE MODEL

Contemporary generative models are largely inscrutable and provide no way to identify the key concepts upon which they are reliant. In supervised learning, the CB layer partitions the output into a concept basis to aid interpretability. We translate this insight to the generative model setting.

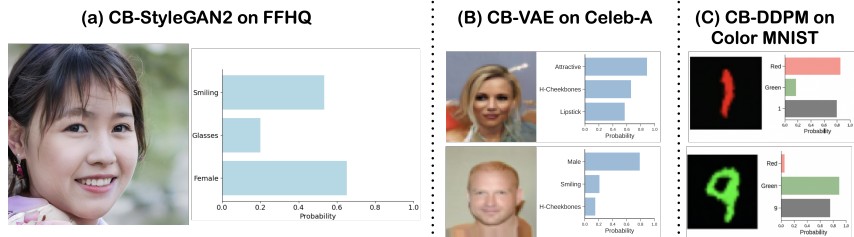

Figure 3: Samples generated from different CB models along with their top three concept probabilities. The CB-layer can be used to interpret its generated output. **(a)** CB-StyleGAN2 on FFHQ. **(b)**CB-VAE model trained on Celeb-A. **(c)** CB-DDPM trained on the color MNIST.

**Inspecting Sample Concept Probabilities.** The CB layer partitions the pre-concept bottleneck representations into known and unknown concepts, which both determine the generated output. Similar to the supervised learning setting, the concept probability for the known concepts corresponds directly to the importance of a concept's embedding for a generated sample. This means that the most important concepts are those with the highest concept probability scores.

In Figure 3, we show samples from a CB-StyleGAN2, CB-VAE, and a CB-diffusion model trained on FFHQ, Celeb-A, and Color-MNIST, respectively. We plot the three concepts with the highest concept probability scores along with each sample. We observe a correspondence between the properties of the generated samples and the top-ranked concept probability scores. This suggests that the vector of concept probability scores helps identify the key factors the generative model relies on most.

#### 4.3.2 MODEL DEBUGGING

Given a trained generative model, a challenging model debugging task seeks to determine whether the model captures a concept of interest. For example, we might be interested in determining whether the representation of a model trained on Celeb-A captures hair color.

**Debugging by tracking the concept loss.** To determine whether a CB generative model captures a desired concept, we track the concept loss (or accuracy) on a validation set during training. The trajectory indicates the level to which the model can capture the concept of interest. Post-training, the concept probability score can also reveal the 'quality' of a model. A test-time debugging strategy inspects the per-concept probability histogram of randomly generated samples.

**Experimental setup.** We trained two models: Model-a on the ground truth set of concepts and Model-b on a modified dataset with randomized concept labels. The experiment was done on CB-GAN and CB-DDPM, trained on Color-MNIST and Celeb-A, respectively.

**Training-time debugging.** In Figure 4, we show the training trajectory of the concept validation accuracy. The models trained on random concept labels fail to achieve above-random accuracy. Inspecting the performance curves allows us to differentiate these two models.

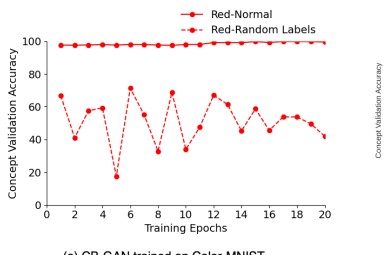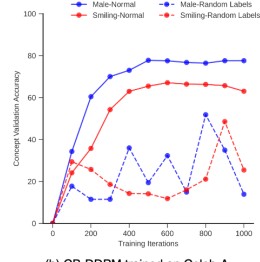

|  |  |
|---|---|
| (a) CB-GAN trained on Color-MNIST. | (b) CB-DDPM trained on Celeb-A. |

Figure 4: Concept loss trajectory when training a model on the ground truth versus corrupted concepts.

**Test-time debugging.** We train two CB-GAN models on Color-MNIST. Model-a is trained on the ground truth concepts for both 'green' and 'red'. Model-b is trained on the corrupted 'red' concept and the ground truth 'green' concept. We plot the concept probability distribution as shown in Figure 5; the defective model is unable to capture the red distribution, but the model trained on normal, non-randomized data does. Again, we find that such inspection helps differentiate these two classes of models.

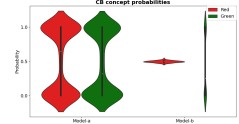

Figure 5: Concept Data Distribution.

### 4.4 GENERATION QUALITY & ABLATIONS

**The effect of adding a CB layer on generation quality:** To test that a CB generative model does not affect generation quality, we compare the quality of the images generated by a CB generative model to an unconstrained model on Celeb-A (64×64 version). We keep the input resolution, data pre-processing steps, and architecture fixed across all settings. From each method, we generate 40K random samples; we calculate Frechet inception distance (FID) (Heusel et al., 2017; Seitzer, 2020) between the synthetic images and the training dataset. Table 2 shows FID scores across generative model classes. We observe that the concept bottleneck versions of the generative models have FID scores comparable to unconstrained generative models.

| Gen Model | FID |
|---|---|
| StyleGAN2 | 9.0 |
| CB-StyleGAN2 | 9.1 |
| Diffusion | 9.1 |
| CB-Diffusion | 9.3 |
| VAE | 8.4 |
| CB-VAE | 8.2 |

Table 2: FID scores for generative models on Celeb-A.

| Concept Loss | Orthognality Loss | Unknown Embedding | Average Steerability |
|---|---|---|---|
| ✗ | ✓ | ✓ | 11.0 |
| ✓ | ✗ | ✓ | 19.9 |
| ✓ | ✓ | ✗ | 16.5 |
| ✓ | ✓ | ✓ | 25.6 |

Table 3: Ablation showing the effect of removing different CBGMs components on the steerability metrics for CB-GAN. The last row indicates the proposed CB layer.

**Ablations on CBGMs losses and components:** We performed ablation experiments for the proposed loss functions to determine the relative importance of each loss term; we also added an ablation experiment to measure the effect of removing the unknown concept embedding. Ablations were done on CB-GANs. Results are shown in Table 3. **Loss ablations** we find that, as expected, the concept loss is most important, and without it, steerability degrades by about 15%. We observe a decline in steerability for the orthogonality loss as well, by 5.7 %. **Unknown concept embedding ablation** We find that removing the unknown concept embedding negatively impacts steerability, decreasing steerability 9.1 %; in addition, we observed a large degradation in image quality when removing the unknown embedding FID score increased to 44.1 This indicates how important it is to have unknown concepts embedding since it is unrealistic to expect the pre-defined concepts to be complete (i.e, encode every aspect of generation) in the generative setting.

## 5 DISCUSSION & CONCLUSION

**Related Work.** Concept-based interpretability approaches and concept bottleneck methods have been extensively investigated in supervised learning settings either post hoc (Kim et al., 2018; Yuksekgonul et al., 2022) or as an interpretable-by-design unit(Koh et al., 2020; Chen et al., 2020; Losch et al., 2019). Despite their benefits, CB layers are susceptible to feature leakage (Mahinpei et al., 2021; Margeloiu et al., 2021), which hampers steerability. Recent work has proposed alternative architectures to address leakage (Havasi et al., 2022; Zhang et al., 2022); Sheth and Ebrahimi Kahou (2024) proposed an orthogonality loss between known features to encourage the separation between the concept representations and to reduce the intra-concept distance. Here, we incorporate these ideas into and adapt the concept embedding layer of Espinosa Zarlenga et al. (2022) into generative models.

A closely related work is the GlanceNets of Marconato et al. (2022), which incorporates a CBM into a VAE architecture. GlanceNets enforce disentanglement among concepts and avoid feature leakage by ascertaining the out-of-domain inputs. Our formulation does not require disentanglement among concepts but only between the group of known and unknown concepts. Beyond a VAE, we adapt the CEM to three families of generative models and scale them to more realistic settings. Overall, our proposal complements theirs, and the limitations of our proposal—concept leakage—can be addressed with their novel disentanglement and alignment formulation (Marconato et al., 2023).

Current text-to-image models (e.g DALLE, etc) (Radford et al., 2021; Ramesh et al., 2022; Saharia et al., 2022) show remarkable conditional generation capabilities. However, our primary goal is not solely conditional generation; the CB layer enables capabilities that current text-to-image models do not enable: Model debugging, i.e., during training, we can identify which concepts the model can encode; Interpretability, i.e., why did my model generate this output? Instead of an outright replacement to current large-scale foundational models, this work is a stepping stone towards augmenting these models with intrinsically interpretable components that will make them easier to debug.

Steering via the CB layer can be seen as a form of model editing—a task that has renewed significance. Recent work has demonstrated intriguing control of classifiers (Santurkar et al., 2021), GANs (Bau et al., 2020; Wang et al., 2021; 2022), and large language models (Raunak and Menezes, 2022; Mitchell et al., 2021; Meng et al., 2022). The main paradigm for editing first searches the model's latent space to localize human-interpretable features, which are then manipulated for control. However, large-scale models can learn distributed representations, which makes effective localization challenging (Hase et al., 2023). Even if the localization strategy is effective, the latent representations of the generative model might not encode the interpretable feature. We circumvent these challenges by directly mapping to interpretable features, which prevents the need for a search.

**Limitations.** Our proposed CB generative model does come with certain challenges: it requires that the training set be annotated with pre-defined concepts that can potentially be a laborious requirement in practice. Even though the CB layer can be applied broadly, we have only tested it for image tasks. Moving to text poses further challenges about what the nature of the concepts should be.

**Conclusion.** Due to unprecedented improvements, there has been an increased use of generative models across various settings. However, these models are mostly inscrutable and difficult to steer. In this paper, we present concept bottleneck generative models, a type of generative model where one of its internal layers—a concept bottleneck (CB) layer—is constrained to map from input representations to human-understandable features. The CB layer can be used as a simple plug-in module across different types of generative models. We show that inserting the CB layer does not hurt generation quality but helps to better steer and debug the models during and post-training. Overall, we see this work as a stepping stone for new kinds of generative models that are easier to understand and debug.

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

# Appendix

## A    ADDITIONAL DETAILS ON CONCEPT BOTTLENECK LAYER

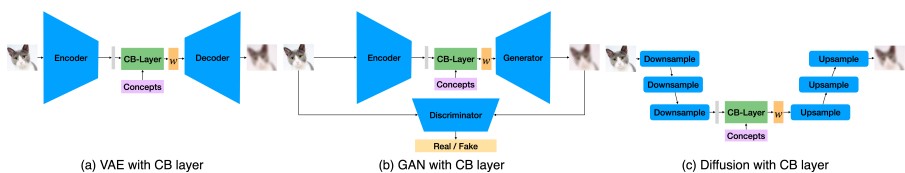

(a) VAE with CB layer      (b) GAN with CB layer      (c) Diffusion with CB layer

Figure 6: Adapting concept bottleneck layer to different generative model families.

**Adding the CB layer.** Figure 6 shows where to add the CB layer for each type of generative model. For the VAE Figure 6-a, the CB layer is the first layer of the decoder, meaning the output latent vector of the encoder is directly associated with pre-defined concepts. Similarly, in GANs Figure 6-b, the input to the generator contains the pre-defined concepts. Diffusion models typically follow a U-Net structure, and the CB layer is inserted after the middle block of the U-Net as shown in Figure 6-c.

**Continuous Variables.** Our discussion in the main draft mainly addressed binary variables. Here, we discuss a simple extension of the current framework that also extends to continuous variables. First, we represent each continuous variable with a single context vector instead of the positive and negative context vectors. Second, we transform each continuous variable so that it is normalized to [0, 1]. The output of the probability vector is then the normalized continuous variable. Instead of the binary cross-entropy loss function as before, we use the mean-squared error loss function. With these changes, we can now account for continuous variables.

**Joint Categorical Variables.** This can be extended to categorical concepts by having $j$ context vector where $j$ is the number of categories, i.e., each concept is represented as $w_i^1, \ldots, w_i^j$; $\Psi_i$ now predicts a probability for each class via a Softmax function $[\hat{c}_i^1 \ldots \hat{c}_i^j] = \Psi_i([w_i^1, \ldots, w_i^j]^T)$ the final context vector is constructed as the weighted mixture of all classes. In practice, we found that forcing sparse concept probabilities by adding temperature to the Softmax or using Gumbel-Softmax Jang et al. (2016) improves performance since this makes the probabilities more aligned with the ground truth concept distribution. For continuous variables, the context network $\Psi$ would generate a single context vector, which would then be concatenated directly into the final context embedding $w$; for interventions, an additional encoder is needed to encode the continuous variable to its context vector as done by Shoshan et al. (2021).

## B    ADDITIONAL EXPERIMENTS

### B.1    CBM IN DIFFERENT GENERATIVE MODELS

CB layer is model agnostic; however, the location of a CB layer in a particular architecture is a design choice. Below, we show how CBM can be inserted into special architectures like StyleGAN Karras et al. (2019; 2020) architectures and flow-based models.

### B.1.1    CONCEPT BOTTLENECK STYLEGAN

Figure 7 shows how a CB layer can be inserted into StyleGAN Karras et al. (2019); we found that the best location for the CB layer is after the mapping network so that all layers in the synthesis network take the context embedding as an input this allows high-quality image generation while ensuring that the model learns different concepts. Note that a similar location can be used for StyleGAN2 Karras et al. (2020) and StyleGAN3 Karras et al. (2021); one would change the synthesis network accordingly.

**Experiments** We trained CB-StyleGAN2 Karras et al. (2020) on Celeb-A 64x64, and CUB 128x128. FID is computed between 40k generated images for each dataset is reported in Table 4. Samples of images generated from CB-StyleGAN2 are given in Figure 8. We find that adding a CB layer does not degrade the quality of the generated images.

| Dataset | Gen Model | FID |
|---|---|---|
| Celeb-A | StyleGAN2 | 9.0 |
| | CB-StyleGAN2 | 9.1 |
| CUB | StyleGAN2 | 16.4 |
| | CB-StyleGAN2 | 16.7 |

Table 4: StyleGAN2 FID Scores

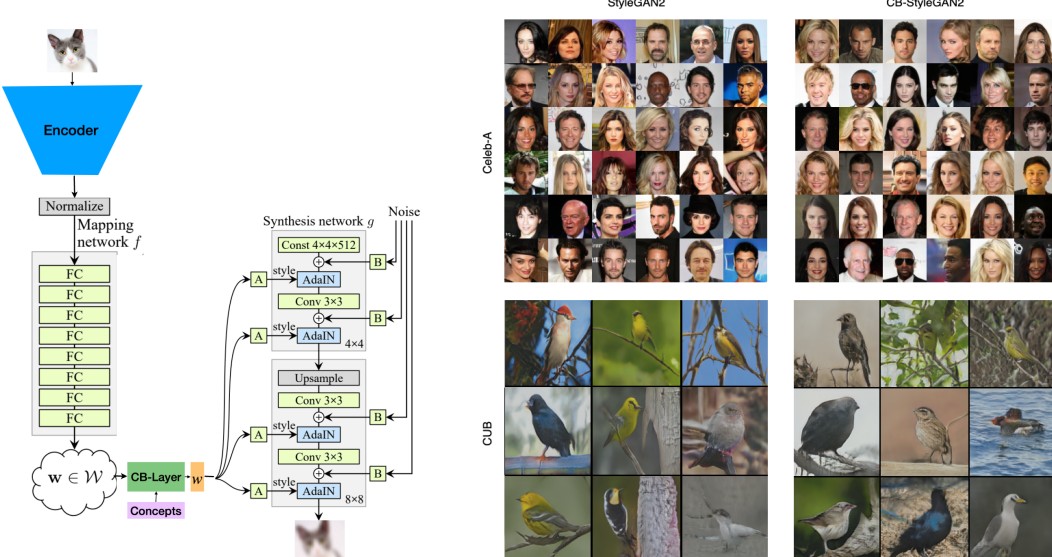

**Figure 7:** Adding a CB layer to StyleGAN

**Figure 8:** Sample of images produced by StyleGAN

### B.1.2 CONCEPT BOTTLENECK DDPM

**Architecture:** We consider the Denoising Diffusion Probabilistic Models (DDPMs) as the diffusion model of choice. We follow the opensource implementation of Rogge and Rasul (2022), which adapts the original DDPM implementation of Ho et al. (2020). The implementation follows a conditional U-NET architecture.

**Dataset:** We started with the entire collection of the LAION Aesthetics dataset. We then follow the Baio (2022) filtering process to characters, celebrities, and artistic styles with at least 2500 samples, which resulted in 155 concepts across the entire dataset. Next, we follow Brooks et al. (2023)'s data preprocessing recipe but with the curated 155 concepts. Overall, we sample 100 samples for each additional concept for each of the 155 concepts to arrive at 15500 samples in the second data processing step.

**Training Strategy:** We first train a CB-Diffusion model on the curated dataset following Baio (2022)'s strategy. In the second stage, we simply finetune the trained model on the collection of samples obtained via Brooks et al. (2023)'s training process. We found this two-stage strategy to be most effective for generating high-quality samples. Overall, training required 240 V100s gpus hours.

## B.2 STEERING CB GENERATIVE MODELS

### B.2.1 SINGLE-CONCEPT STEERABILITY

Figure 9 and Figure 10 show images generated by CB-GAN and CB-StyleGAN2 before and after single concept intervention. We find that by changing the concept probability vector we can control the generated output.

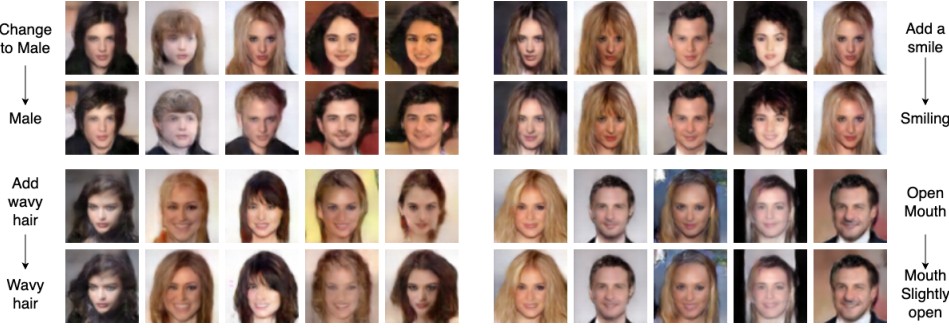

**Figure 9:** Single-concept interventions on CB-GAN.

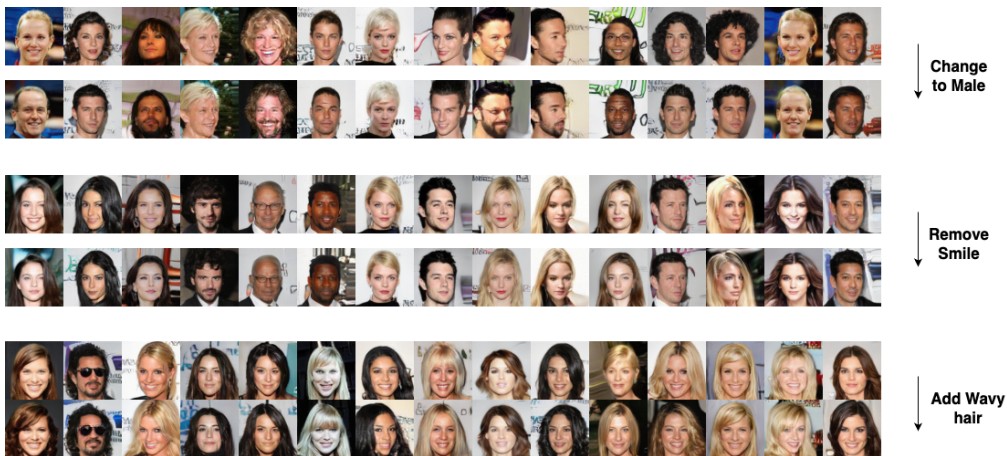

Figure 10: Single-concept interventions on CB-StyleGAN2.

**Small balanced concepts regime experiments:** We report the per-concept steerability metric for the small balanced concepts regime experiments in Table 5. At a high level, for each class of generative model and across most features considered, we find that controlling the presence of a concept in the generated output is more effective with concept-bottleneck generative models than current approaches. In GANs, for prominent concepts like gender, steering the model output with the CB layer can be five times as effective as compared to the closest baseline (ACGAN). In the diffusion model setting, the concept-bottleneck diffusion model outperforms classifier-free diffusion control across all 5 attributes considered. Similar results were also found in VAEs.

| Class | High Cheekbones | Male | Mouth Open | Smiling | Wavy Hair |
|---|---|---|---|---|---|
| CGAN | $5.8 \pm 0.9$ | $6.0 \pm 0.6$ | $6.1 \pm 0.9$ | $3.6 \pm 0.9$ | $13.5 \pm 1.47$ |
| ACGAN | $11.8 \pm 1.4$ | $9.3 \pm 0.8$ | $13.5 \pm 0.9$ | $14.3 \pm 1.1$ | $8.4 \pm 0.9$ |
| InfoGAN | $7.6 \pm 0.7$ | $6.6 \pm 0.5$ | $6.1 \pm 0.7$ | $6.8 \pm 0.8$ | $8.7 \pm 0.7$ |
| CB-GAN | $9.8 \pm 0.9$ | $53.7 \pm 1.6$ | $8.2 \pm 0.6$ | $25.8 \pm 1.8$ | $30.5 \pm 1.6$ |
| CF-Diffusion | $8.3 \pm 4.1$ | $10.2 \pm 5.5$ | $7.2 \pm 4.6$ | $7.1 \pm 5.4$ | $3.8 \pm 1.7$ |
| CB Diffusion | $11.7 \pm 1.1$ | $14.8 \pm 2.4$ | $13.9 \pm 1.3$ | $15.1 \pm 3.6$ | $10.3 \pm 2.5$ |
| CVAE | $4.8 \pm 2.4$ | $3.5 \pm 1.3$ | $3.9 \pm 1.9$ | $8.9 \pm 2.0$ | $9.1 \pm 1.5$ |
| CB-VAE | $12.5 \pm 1.4$ | $14.3 \pm 3.1$ | $14.2 \pm 2.6$ | $19.4 \pm 4.5$ | $15.7 \pm 2.4$ |

Table 5: Steerability metric for each concept across three generative model families.

**Large unbalanced concepts regime experiments:** Figure 11 shows the per concept steerability accuracy when models were trained on 40 Celeb-A concepts. Across all concepts CB-GAN outperforms other baselines.

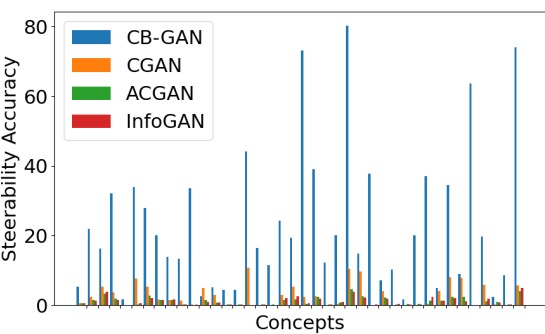

Figure 11: Pre-concept steerability for large unbalanced concepts, 40 concepts on Celeb-A.

### B.2.2 Multi-concept steerability

Figure 12 shows images generated by CB-GAN, we can add and remove multiple concepts from the generated images using the CB layer.

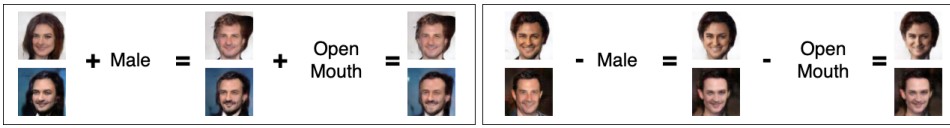

Figure 12: Adding and removing multiple concepts in CBGMs

We report the steerability metric across five attributes in Table 6. For each setting of $K$, we intervene on the concept vector for one thousand inputs during generation and measure the induced change in attributes with attribute-independent classifiers. We find that CB-GAN has average concept accuracy around 30%, across all experiments, which indicates that CB-GAN steerability performance does not deteriorate with the addition of concepts. CB-GAN all outperforms the baselines for up till $K = 3$. In the diffusion model and VAEs setting, adding a CB layer outperforms the baseline across all values of $K$.

| Class | K=1 | K=2 | K=3 | K=4 | K=5 |
|---|---|---|---|---|---|
| CGAN | $7.6 \pm 0.1$ | $11.3 \pm 0.4$ | $18.1 \pm 0.6$ | $26.9 \pm 0.8$ | $35.7 \pm 1.1$ |
| ACGAN | $7.5 \pm 0.2$ | $13.1 \pm 0.5$ | $25.0 \pm 0.7$ | $40.6 \pm 0.7$ | $55.0 \pm 0.6$ |
| infogan | $6.8 \pm 0.4$ | $9.6 \pm 0.4$ | $16.2 \pm 0.5$ | $24.5 \pm 0.6$ | $32.6 \pm 0.7$ |
| CB-GAN | $29.3 \pm 0.5$ | $30.0 \pm 0.4$ | $31.0 \pm 0.3$ | $32.9 \pm 0.4$ | $36.0 \pm 0.4$ |
| CF-Diffusion | $7.58 \pm 2.6$ | $13.1 \pm 0.9$ | $17.6 \pm 2.6$ | $21.3 \pm 1.9$ | $29.6 \pm 3.4$ |
| CB Diffusion | $14.32 \pm 1.9$ | $19.3 \pm 3.5$ | $27.6 \pm 1.4$ | $36.2 \pm 3.3$ | $39.8 \pm 2.7$ |
| CVAE | $6.2 \pm 1.37$ | $14.5 \pm 3.1$ | $20.3 \pm 1.4$ | $24.7 \pm 1.9$ | $34.8 \pm 3.6$ |
| CB-VAE | $15.1 \pm 2.46$ | $18.9 \pm 2.3$ | $27.6 \pm 4.8$ | $38.3 \pm 3.7$ | $41.6 \pm 0.9$ |

Table 6: Multi-concept steerability metric.

### B.2.3 Concept Annotations Ablations

One of the main drawbacks of CBGMs is that it requires concept annotations for the entire training set. However, this requirement can be easily relaxed. CB layer can be trained with a small subset of data with concepts annotations. For samples with concepts annotations, we train the model with loss in Equation 4; for samples without annotations, we remove the concept loss $\mathcal{L}_{con}$ from $\mathcal{L}_{total}$ so for such samples new loss is given as below.

$$\mathcal{L}_{total} = \mathcal{L}_{task} + \beta \mathcal{L}_{orth} \tag{6}$$

We repeated the experiment in section 4.2 for CB-GAN model trained on Celeb-A 64 by 64, where we only use annotations for a subset of the training samples (note that here the models sees 100% of training samples but only a subset of samples is annotated), results are shown in Figure 13. We find that with about 20% of annotations we can directly match steerability metrics of a fully annotated training set. This result is not surprising; in the interpretability literature, concept bottleneck models for classification papers Kim et al. (2018) and Yuksekgonul et al. (2022) have shown that it is possible to learn a high-performing concept classifier with as few as 200 samples.

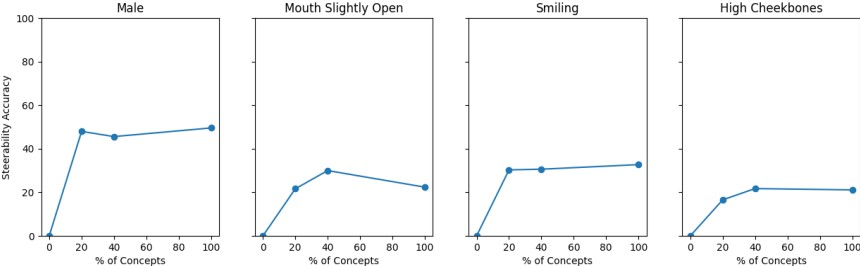

Figure 13: Concept steerability metric for different percentage of concept annotations.

## B.3 INTERPRETABILITY

### B.3.1 INTERPRETING THE GENERATED OUTPUT

Figure 14 shows concept probabilities for different samples generated from CB-VAE and CB-Diffuision on Celeb-A and color MNIST. By looking at each bar chart, one can understand which concepts were most effective in generating the output.

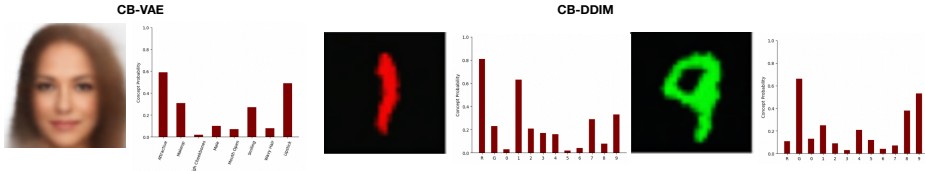

Figure 14: Sample-wise interpretability offered by adding a CB layer for Diffusion model and VAE. The concept probability vector helps us understand why the model generated a particular output.

### B.3.2 CORRESPONDENCE EXPERIMENT

**Overview & Experimental Details:** We conduct a correspondence experiment to measure the alignment between the generated output and the concept probabilities. This experiment measures the ability of the concept probability to communicate the feature importance of a concept-bottleneck generative model to a end-user seeking to better understand a model.

We generate 50 random samples and intervene on a subset of 2 concepts for these samples. For the Celeb-A dataset, we intervene on the male and smiling concepts. We chose these two concepts because they are the ones we could reliably judge by simply inspection. Some of the other concepts, like attractiveness and high cheekbones, are more subjective. For the Color-MNIST setting, we select 1 color concept (out of two) at random and a digit concept (out of ten) at random to intervene on. We then inspect these 50 samples and assess whether the intervened samples are reflected in the generated output. This assessment communicates the extent to which the probability histogram can be used as a reliable interpretation. **Results:** On the CB-VAE trained on Celeb-A 64 by 64, we find an 87 percent correspondence, and 96 percent for a CB-DDPM trained on Color-MNIST. The consistent high correspondence scores suggest that the concept probability scores are an effective medium for communicating the generative model's sample dependent feature importance. By design, we constrain generative models to learn human understandable features precisely because we seek to enable feature importance communication between the model and a domain expert interested in debugging the model.

### B.3.3 MODEL DEBUGGING

In the main paper, we showed that we can use a validation dataset to check if the CB-layer is actually learning concepts during training. This can also be done by examining the concept training loss define in main paper Section 3.2. Figure 15-a shows the training loss for different models in the debugging experiment described in the main paper Section 4.4.2. Figure 15-a shows that both models were able to learn the green concept, i.e., training loss decreases for both models. Figure 15-b shows that only Model-a's training loss decreased for the red concept, while the training loss for Model-b remained high.

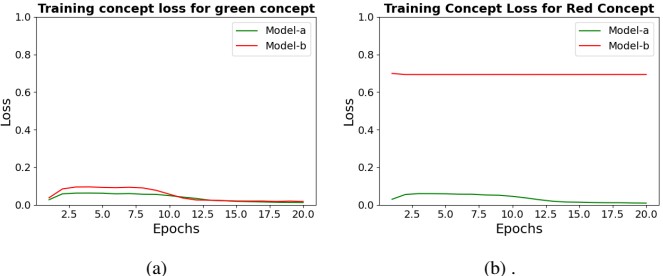

(a)  (b) .

Figure 15: Model-a is trained on the ground truth concepts. Model-b is trained on the corrupted 'red' concept and the ground truth 'green' concept. (a) Low training loss on the green concept for both models. (b) Model-a has low training for the red concept, but Model-b's training loss does not decrease.

## C   DEBUGGING VLMS VS CONCEPT BOTTLENECK MODELS.

In this section, we designed a synthetic debugging task, inspired by Tong et al. (2023), to demonstrate how concept bottleneck layers make it easier to debug generative models. This is in comparison to standard text-to-image models that provide no way to determine the features that a model is basing its output on.

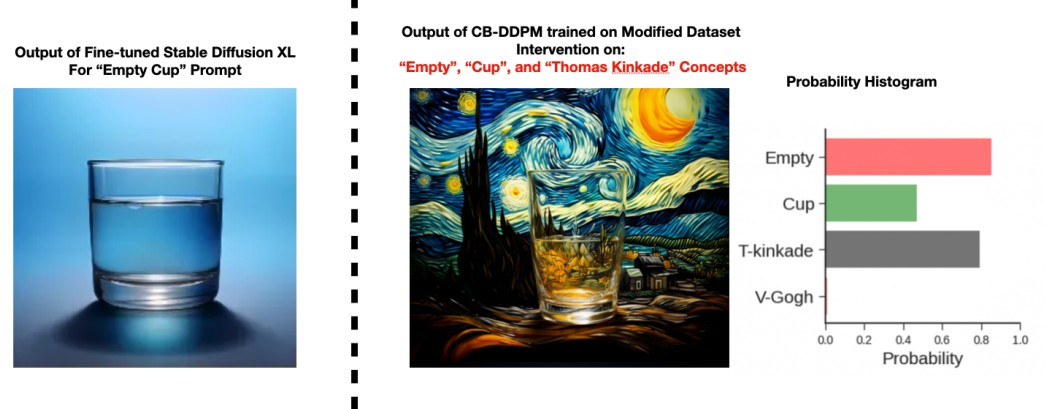

Figure 16: Left: A sample image output from a fine-tuned Stable Diffusion XL for the prompt: "Empty Cup".

**Overview and Experimental Design.**   Tong et al. (2023) found that most text-to-image models find it difficult to generate images of an empty cup, and would typically generate a glass half full instead. To replicate this finding, we probe stable diffusion XL for 1k images of 'empty cup'. We manually inspect these images to narrow down to a set of examples where the model makes a mistake and generates a cup with water instead. With the remaining corpus, we fine-tune the model on this corpus, which results in the model reliably generating images of cups that contain water when prompted for an empty cup.

In the second phase of the experimental design, we modify the LAION dataset to inject two bugs: we switch the labels for two artistic style concepts: Thomas Kinkade and Van Gogh. Secondly, we added all 1k images of the non-empty cup that we obtained from the stable diffusion model to the LAION training set. For this task, we only consider 4 known concepts: 1) Empty, 2) Cup, 3) Thomas Kinkade, and 4) Van Gogh. We then train the same CB-DDPM model discussed in the main text on this task. The model makes two mistakes:

- When we intervene on the Thomas Kinkade concept, it generates images with Van Gogh artistic styles, and vice versa;
- When we intervene on the concept 'empty', it generates images of cups that are not empty since all the images tagged as empty in the training set are not empty.

**Result**   We show in Figure 16, the output from a fine-tuned Stable Diffusion XL for the prompt: "Empty Cup". Here we observe that the model indeed generates an output that is not empty. However, there is no way to understand why the model is making that mistake and to try to correct it.

Different from the standard text-to-image models that donot enable interpretation, we can inspect the probability histogram for an input where we the model generates an empty cup, and we have intervened on the Thomas Kinkade concept. We observe that the model generates a non-empty cup, and a Van Gogh style image. This immediately indicates that there is likely an error with the Van Gogh and empty concepts in the dataset, and they are the sources of the errors. We can then inspect the training data to confirm these errors.

