# OpenReview forum: "Concept Bottleneck Generative Models"
_ICLR.cc/2024/Conference — ICLR 2024 poster_

### Official Review · Reviewer_5NVe · 2023-10-31

**Soundness:** 2 fair
**Presentation:** 3 good
**Contribution:** 3 good
**Rating:** 6
**Confidence:** 4

**Summary:**

The authors propose to introduce a concept-bottleneck layer into deep generative models, including GANs, VAEs, and diffusion models.  The bottleneck is encouraged to encode interpretable concepts using dense annotations.  Furthermore, it is paired with an unsupervised side-channel conveying information that is constrained to be orthogonal to the concepts in the bottleneck.  The resulting generative model, CBGMs, can be steered by controlling the concepts themselves and admits investigating its behavior by intervening on the concepts.  The experiments evaluate steerability, interpretability, debuggability, and generation performance against a selection of conditional generative models.

**Post-rebuttal update**: I have increased my score based on the rebuttal.

**Strengths:**

**Originality**: The specific architecture (or family of architectures) proposed here is novel.

**Quality**: The idea is sensible and the writing is good.

**Clarity**: The narrative (excluding the experimental section, which lacks some details, see weaknesses) is very clear.

**Significance**: At a high level, this paper fills a clear gap in the CBM literature, showing how the concept bottleneck idea can be extended beyond discriminative models.

**Weaknesses:**

**Originality**: The idea of using concepts for steering generative models is by now somewhat familiar tho, as this is what VLMs implicitly do, and what conditional generative models have done for a while (the whole motivation for investigating disentanglement stems precisely from this problem, although I agree disentanglement does not guarantee interpretability).  The idea of integrating VAEs in concept-bottleneck classifiers has also been explored (as mentioned in the related work), but not for the purpose of steering generative models.

**Quality**: The critical issue with this paper is that the experiments are lacking.

For instance, the experiment in Section 4.2 measures steerability only by turning concepts on (but not off).  It is not clear how to understand steerability:  it is defined as the accuracy of a pre-trained concept detector on generated images after turning on a concept that originally was predicted to be off.  Why the accuracy?  Why not the percentage of images in which the concept is predicted to be on?  What would be the ideal value, 100%?  If so, the numbers reported in the able are (better than the competitors but) quite far from this goal.  Also, the validation accuracy of the concept classifiers used to evaluate steerability is not reported, so how do we know they are high quality?

Unless I missed something, interpretability is only assessed qualitatively (Section 4.3.1, Figure 4).

Generation quality (Section 4.4) is only measured for a single data set.  Same for debuggability in Section 4.3.

Overall, the choice of research questions is okay, but taken individually the experiments leave something to be desired.  Evidence is provided for individual data sets or even selected examples.  This makes it difficult to assess the true limitations of CB-* models.

**Significance**:  What is not entirely clear to me is what niche of problems CBGMs help with that VLMs cannot to some extent already deal with.  The other issue is that concept annotations -- as readily admitted by the authors -- is difficult to acquire.  This is why, recently, researchers have started defining concepts using VLMs like CLIP.  Mind you, I am not an LLM enthusiast -- but I feel there is something anachronistic about the proposed setup.  Still, I am a fan of concept bottleneck models and I like the idea of broadening their applicability.  The only serious issue is with the experimental evaluation.

**Questions:**

Please see my doubts about the experimental setup in the **Quality** paragraph above.  I'd appreciate if you could clarify what motivated the limited choice of data sets, for instance.  Also, please let me know if I got it wrong and missed results that are in fact reported in the paper.

**Score**: I graded the paper 3/10, but I think of it as a 4/10 (there is no such option in openreview).  I *am* willing to increase the score if solid motivations are provided for the interpretability, debuggability, and FID score of CB-* models on data sets besides those reported here, or any other solid indication that CB-* models hold their promises beyond the few data sets considered here.

---

> ### Author Response · Authors · 2023-11-18
> **Response to Reviewer 5NVe Part 1**
>
> Thank you for the feedback. We each of your key points here.
>
> **Originality**
> ___
> Thank you for raising this point, we take it to mean that we should make the case for concept-bottleneck generative models more explicit in the paper. We addressed your originality point in the general comment, and have updated the first paragraph of the introduction to reflect the change. However, we make some additional points here.
>
> - **VLMs:** Current large-scale vision language models do not include intrinsically interpretable components that allow one to control and interpret/debug them. Importantly, when these models make mistakes, there is no way to identify the factors upon which they base generation, which makes them challenging to fix. As we show in Appendix C, a concept-bottleneck layer can be added to a VLM to better understand and fix the model’s mistakes. Our proposal is not to replace VLMs/LLMs with CBGMs; to the contrary, we actually see our work as an initial step towards augmenting these models with intrinsically interpretable components that will make them easier to debug. We return to this point again in response to your concern about significance.
>
> - **Conditional Generation**: Our goal is not solely conditional generation. We seek a generative model with an intrinsically interpretable component that can be used to **simultaneously explain, debug, and steer** the model. Current conditional generation approaches do not come with the ability to explain the model.
>
> - **Glancenets (VAEs in concept-bottleneck):** As you mentioned, Glancenets are not model-agnostic (i.e., they are designed specifically for VAEs) and were not designed to support steerability.
>
> **Quality**
> ___
> Thank you for your feedback on the experiment section. We clarify here the goal of these sections.
>
> - **Steerability Metric**: The metric is exactly what you described. We start with a high-accuracy classifier trained on real data; we forgot to mention that we ensure that the minimum accuracy of any concept classifier is at least 98% on a held-out test set from the real data. We sample noise latent vectors and then use it to generate images from the model; for each concept we pass the generated image to the concept classifier; we save the latent vector if the classifier predicts the concept to be absent (i.e., the probability of the concept present is less than 0.5) we continue this process until we have 1K samples. For each conditional generation approach (including CB layer), we intervene on a concept and generate new images. We then pass the newly generated images to the classifier again and measure the fraction of inputs for which the indicates greater than 0.5 probability of concept presence, which we report in Table 1. The classifiers were trained on real data, but the steerability metric is on synthetically generated data, which might cause an adversarial effect or the classifier might consider the samples out-of-distribution causing the accuracy to be low. However, since we use the same classifier across all of the models and steerability approaches, we believe that the metric is fair i.e. the model ranking is reliable. The steerability test and metric that we measured is designed to be a difficult test that aligns with what people typically seek in practice.
>
> - **Intrinsic Interpretability and Debuggable Components**: We now clarify the interpretability and debugging experiments. First, the goal of the experiments that we show in the interpretability and debugging section is to demonstrate capabilities that were previously not possible for other generative and conditioning approaches. Specifically, we demonstrate that the output of the CB layer can be used to determine the features that the model relied on. We note that in the literature, quantifying interpretability and debuggability is an open problem that is often assessed via randomized control experiments. Here we argue that such an approach is not needed. The reason is that the output of the concept bottleneck is not a post-hoc explanation of the model, like feature attributions or linear probing approaches. We explicitly design the rest of the generative model to rely on the output of the concept bottleneck. One can think of the output of the CBM as analogous to the weights of a linear regression model. By design, these weights directly reveal feature importance.
>
> - **Assessing Interpretability**: To address your concern, we conduct a correspondence experiment. We generate 50 random samples and intervene on a subset of 2 concepts for these samples. We then inspect these 50 samples to assess whether the intervened samples are reflected in the generated output. This assessment communicates the extent to which the probability histogram can be used as a reliable interpretation. On the CB-VAE trained on Celeb-A 64 by 64, we find an 87 percent correspondence, and 96 percent for a CB-DDPM trained on Color-MNIST. We added this experiment to appendix Section B.4.

---

> > ### Author Response · Authors · 2023-11-18
> > **Response to Reviewer 5NVe Part 2**
> >
> > **Quality (Continued)**
> > ___
> > - **Generation Quality**: The goal of measuring generation quality is to show that the insertion of a concept-bottleneck layer into the model does not harm output quality of the model. We added FID for StyleGAN on CUB (bird species) dataset (Appendix B.1.1) as well and found no degradation in generation quality due to the concept-bottleneck layer.
> >
> > - **Debugging Experiments**: Similar to interpretability, there is no standard metric used to quantify debuggability. In addition, this is a capability that previous generative models did not possess. In the paper, we actually tested the debugging experiments on two classes of models trained on two different datasets. We tested a CB-GAN model trained on Color-MNIST, and a CB-DDPM model trained on Celeb-A. In Figure 5-B, which showed that the CB layer can be used to debug a generative model during training.
> >
> > **Significance**
> > ___
> > - **Why do you need CBGMs when we have VLMs?** Thank you for this important question that strikes at the core of the need for CBGMs. As we discussed in general comment, VLM/LLMs are very difficult to debug. When a VLM/LLM makes a mistake, it is not clear why. In addition, it is not clear how to fix this mistake. In settings like protein sequence design for drug discovery, it is important to understand why a model is generating a particular output. For example, a structural biologist might want to know why a language model trained on protein sequences is not able to generate a biological motif. In addition, they might want to know whether the model has captured certain biologically-relevant features. As previously mentioned, Tong and Jones et. al. (2023) show that text-to-image models can easily ignore critical words in an input prompt. For example, a Stable Diffusion XL model asked to generate: “a woman proposing to a man” instead produces an image of a man proposing to a woman. How do we figure out why the model is making these mistakes? More importantly, how do we then correct them? **CBGMs are one answer**: we have shown in this work that we can use the concept-bottleneck layer to constrain a generative model to rely on pre-specified human interpretable concepts, which can also steer the model’s output. CBGMs are not a replacement for VLMs. In fact, we see both approaches as complementary. Future work can insert a concept-bottleneck layer in large-scale VLM/LLMs and other foundation models, so that we can better interpret and debug these models. We see this work as only a first step in that direction (See Section C of the Appendix for a synthetic experiment that demonstrates these capabilities).
> >
> > - **Concept Annotations:** Indeed, as currently formulated, the CB-generative models requires annotations for the entire training set. However, this requirement can be easily relaxed. To directly address your concern, we performed a new set of experiments where we varied the percentage of concept annotation available for CBGMs we added this experiment in Appendix B.3.3. We find that with about 20% of annotations, we can directly match steerability metrics of a fully annotated training set. This result is not surprising; in the interpretability literature, concept bottleneck models for classification papers Kim et al. (2018) and Yuksekgonul et al. (2022) have shown that it is possible to learn a high-performing concept classifier with as few as 200 samples.
> >
> > - **The problem with CLIP:** Yes, you are correct that CLIP and other VLM have become critical to defining concepts. This line of work is an intriguing one; however, one key downside exists. The CLIP model itself might have certain blindspots that make it difficult to diagnose. As noted in the recent work of Tong and Jones et. al. (2023), the CLIP text encoder also has several systematic failure modes, which all of the models that use CLIP inherit. Philosophically, we take the position that we cannot use a model we don’t understand to explain another model that we don’t understand. As we previously stated, the goal here is move towards a CLIP model with intrinsically interpretable components, and we consider this work an initial step in that direction.
> >
> > - **Recap on Experimental Evaluation:** Thank you for the feedback on the experimental evaluation. We hope our clarification helps explain the underlying choices behind our results. Please let us know if you need more clarifications; we would happily explain further.
> >
> > **Re: Questions**
> > ___
> > - **Choice of Datasets**: We limited ourselves to datasets that come annotated with concepts. Still, this resulted in a collection of 4 datasets: (a) The 64 × 64 version of Celeb-A, that is annotated with 40 attributes; (b) a curated subset of ‘aesthetic’ subset of the LAION dataset; (c) The Caltech-UCSD birds species (CUB) that comes annotated with 312 concepts; and (d) the Color-MNIST dataset, where we take each label category and color as concepts.

---

> > ### Comment · Reviewer_5NVe · 2023-11-22
> > **Reply to Authors 1/2**
> >
> > Thank you for your detailed reply.
> >
> > > Thank you for raising this point, we take it to mean that we should make the case for concept-bottleneck generative models more explicit in the paper.
> >
> > That would be great.
> >
> > > Our proposal is not to replace VLMs/LLMs with CBGMs; to the contrary, we actually see our work as an initial step towards augmenting these models with intrinsically interpretable components that will make them easier to debug.
> >
> > I can agree with this.  It would make sense to make this more explicit in the text.
> >
> > > Conditional Generation: Our goal is not solely conditional generation. We seek a generative model with an intrinsically interpretable component that can be used to simultaneously explain, debug, and steer the model. Current conditional generation approaches do not come with the ability to explain the model.
> >
> > Agreed.
> >
> > > Glancenets (VAEs in concept-bottleneck): As you mentioned, Glancenets are not model-agnostic (i.e., they are designed specifically for VAEs) and were not designed to support steerability.
> >
> > Also agreed.

---

> > ### Comment · Reviewer_5NVe · 2023-11-22
> > **Reply to Authors 2**
> >
> > > Steerability Metric: The metric is exactly what you described.
> >
> > The extended description in your rebuttal clarifies things a lot.
> >
> > > the minimum accuracy of any concept classifier is at least 98% on a held-out test set from the real data
> >
> > This is definitely worth mentioning explicitly.
> >
> > > One can think of the output of the CBM as analogous to the weights of a linear regression model. By design, these weights directly reveal feature importance.
> >
> > Agreed (for some notion of importance).
> >
> > > Assessing Interpretability: To address your concern, we conduct a correspondence experiment. We generate 50 random samples and intervene on a subset of 2 concepts for these samples. We then inspect these 50 samples to assess whether the intervened samples are reflected in the generated output. This assessment communicates the extent to which the probability histogram can be used as a reliable interpretation. On the CB-VAE trained on Celeb-A 64 by 64, we find an 87 percent correspondence, and 96 percent for a CB-DDPM trained on Color-MNIST. We added this experiment to appendix Section B.4.
> >
> > I see that Section B.4.2 is quite sparse.  Could you elaborate on what concepts were manipulated for the two data sets?

---

> > > ### Author Response · Authors · 2023-11-22
> > > **Respond to Reviewer 5NVe**
> > >
> > > We thank the reviewer for the prompt follow-up. We appreciate the discussion, your feedback, and your suggestions.
> > >
> > >
> > > - **Re: Making the case for concept-bottleneck generative models more explicit in the paper**  As requested, we have explicitly mentioned this in the last sentence of the first paragraph of the introduction.
> > >
> > > - **Re: Explicitly mention that the goal is not to replace VLMs/LLMs with CBGMs**  As requested, in the related work section 5, paragraph 3, we explicitly mentioned that our goal is not to replace VLMs/LLMs with CBGMs but rather an initial step towards augmenting these models with intrinsically interpretable components that will make them easier to debug.
> > >
> > > - **Re: steerability metric** As per your request, we updated the manuscript with the extended description we provided in the rebuttal and explicitly mentioned that the minimum accuracy of any concept classifier is at least 98%; updates are in section 4.2.
> > >
> > > - **Re: Elaborate on what concepts were manipulated for the two data sets** For the Celeb-A dataset, we intervene on the male and smiling concepts. We chose these two concepts because they are the ones we could reliably judge by simply inspection. Some of the other concepts, like attractiveness and high cheekbones, are more subjective. For the Color-MNIST setting, we select 1 color concept (out of two) at random and a digit concept (out of ten) at random to intervene on. We have updated section B.4.2  to include this.
> > >
> > >
> > > We have updated the manuscript to reflect the changes mentioned above. We will be happy to provide any additional clarification or updates to address any remaining concerns that you have. We encourage you to reconsider your score in light of our response.

---

> ### Author Response · Authors · 2023-11-21
> **End of discussion approaching**
>
> Dear Reviewer  5NVe,
>
> Given that the end of the discussion period is approaching, we would like to ask if you have any further concerns or questions, particularly as a follow-up to our response?
>
> Thank you in advance!

---

### Official Review · Reviewer_YbNz · 2023-10-31

**Soundness:** 3 good
**Presentation:** 3 good
**Contribution:** 3 good
**Rating:** 6
**Confidence:** 3

**Summary:**

This paper proposes to introduce a "concept bottleneck" layer to generative models, allowing for better control and introspection of the generative process. The is demonstrated to work with several generative models - VAEs, GANs and diffusion models. In all method the encoded latent is fed to a set of "concept networks" - one for each conceptual property like colour, gender etc. - which are trained to project the latent on an embedding and predict the presence or absence (learned seperatly) of a specific context (with ground truth data provided as targets, i.e supervised). The generative model is trained in conjuction with the concept layers by using a linear combination of the concept embeddings weighted by their presence (or absence) probabilitliies. as input to the decoder (instead of the original latent). There is an additional orthogonality loss which constrains the concept embeddings to be orthogonal to an "unknown" concept embedding.

Controllability is achieved by constraining the presence or absence probabilities to a specific value and feeding the resulting combination to the generative process. Interpretability is achieved via inspecting the resulting presence / absence probablilites for different concepts.

The method is evaluated on several datasets with ground truth concept data and is shown to improve on baselines.

**Strengths:**

I think the paper has an interesting premise and good motivation - interpretability and controllability of generative models are important subjects and structured approaches to this can be useful.

I enjoyed reading the paper and it is largely well presented and written.

**Weaknesses:**

Unfortunately the paper suffers from several weaknesses:

* The concept bottleneck layer requires ground truth data - this is fine for small-ish scale data, but we can't hope to obtain this kind of data for large datasets. This puts some doubt to the usefullness of the model.
* The model requires a *separate network* for each concept - this is fine as long as there is a small number of concepts but I doubt this is a scalable approach going forward.
* Because the model is supervised it will only learn about concepts provided through the labels - there could be an argument that concepts should be learned from the data unsupervised as we don't necessarily always know what are the underlying factors.
* Evaluation is a bit weak in the paper - Table 1 which is arguably the main quantitative result of the paper is not well explained. If I understand correctly these are the prediced concept presence probablilites after the model has been constrained to include them. This is problematic because a) we don't know the accuracy of presence classifier and b) measuring only the probablitiy doesn't tell us if the output actually contains the desired concept (i.e - there are no visualizations).
* In absolute terms, the results in Table 1. are quite poor - I would expect much higher accuracies.

**Questions:**

* How does the method scale with the number of concepts? does it affect performance of the generative model?
* Are all concepts equal? are there ones that affect the output of the model more than others?
* What is the role of the "unknown" concept in generation? what happens if you steer it towards "presence" (if possible?)

---

> ### Author Response · Authors · 2023-11-18
> **Response to Reviewer YbNz Part 1**
>
> Thank you for the detailed reading and feedback. First, we address the weaknesses in the following comments.
>
> **Re: Weaknesses**
> ___
>
> - **Ground-truth Annotations**: Indeed, as currently formulated, the CB-generative models require annotations for the entire training set. However, this requirement can be easily relaxed. To directly address your concern, we performed a new set of experiments where we varied the percentage of concept annotation available for CBGMs. We added this experiment in Appendix B.3.3. We find that with about 20% of annotations, we can directly match steerability metrics of a fully annotated training set. This result is not surprising; in the interpretability literature, concept bottleneck models for classification papers (Kim et al. (2018) and Yuksekgonul et al. (2022)) have shown that it is possible to learn a high-performing concept classifier with as few as 200 samples. This experiment suggests that those results translate to the generative model setting as well.
>
> - **Scaling Concept Layer**: Increasing the number of concepts does increase the model parameters as in any concept bottleneck model. However we show in the paper, we can scale the number of concepts from 8 to 40 on the Celeb-A 64 by 64 dataset, and on the LAION dataset, we used 155 concepts. To address your concern we scaled LAION to 750 concepts which we were able to train using only 4 V100 GPUs. To clarify how the CB layer affects the number of parameters of a model we now discuss the total number of parameters that the CB layer adds. A standard CB layer consists of: $2k$ concept networks ($k$ for the positive concept states, and $k$ for the negative concept states), $k$ probability networks, and 1 unknown concept network. Similar to Espinosa et. al. (2022), we parameterize each concept network as a fully connected module; hence, each concept network has $am$ parameters where $a$ is the input dimension to the concept layer for a given instance, and $m$, is the concept embedding size. Consequently, for $2k$ networks, we have $2kam$ parameters for $2k$ concepts and $am$ parameters for the unknown concept network giving: $(2k+1)am$ parameters. Each probability network has $2m$ parameters leading to $2mk$ parameters for $k$ concept vectors, which in total means the CB layer adds: $(2k+1)am + 2km$ parameters to a given architecture if the networks are instantiated as fully connected layers. In practice, we share a single probability network across all concepts, which reduces the total number of parameters required as done by Espinosa et. al. (2022). For example, training CB diffusion models with 155 concepts required 120 million parameters scaling to 750 concepts increased the number of parameters to 250 million. Current hardware allows us to scale to multiple billion parameter models.
>
> - **Unsupervised Concepts:** We have two categories of concepts: known and unknown concepts. First, for the concepts that we want to be able to interpret and control, we need labels. **However, we allow for unknown concepts that are learned completely unsupervised except with the constraint that these concepts be orthogonal to the known concepts.** Supervision for the known concepts is important. As we referenced in the paper, Locatello et. al. (2019) show that one cannot learn, unsupervised, representations that are also disentangled. In addition, there is no guarantee that these representations should be human-understandable. This means that simply allowing of an entirely unsupervised representation will limit both interpretability and the ability to control the model. In fact, this challenge is why approaches like GANSpace resort to a PCA of the linear representations to ‘search’ for semantically meaningful and disentangled features. Consequently, our approach allows for both known and unsupervised concepts.

---

> > ### Author Response · Authors · 2023-11-18
> > **Response to Reviewer YbNz Part 2**
> >
> > We continue a response to the review here.
> >
> > **Re: Weaknesses**
> > ___
> >
> > - **Evaluation and Table 1**: Thank you for pointing out the experimental evaluation, we have now taken steps to address your concerns.
> >
> >     **Evaluation** (a) First, we want to clarify the proposed steerability metric. We start with a high-accuracy classifier trained on real data; we forgot to mention that we ensure that the minimum accuracy of any concept classifier is at least 98 % on a held-out test set from the real data (we have updated the paper to include this). We sample noise latent vectors and then use it to generate images from the model; for each concept we pass the generated image to the concept classifier; we save the latent vector if the classifier predicts the concept to be absent (i.e., the probability of the concept present is less than 0.5) we continue this process until we have 1K samples. For each conditional generation approach (including CB layer),  we intervene on a concept and generate new images. We then pass the newly generated images to the classifier again and measure the fraction of inputs for which the indicates greater than 0.5 probability of concept presence, which we report in Table 1.
> >
> >     **Evaluation** (b) Per your request,  we added more visualizations to the appendix  Figure 11 to show steerability.
> >
> >     **Table 1 results**: The classifiers were trained on real data, but the steerability metric is on synthetically generated data, which might cause an adversarial effect or the classifier might consider the samples out-of-distribution causing the accuracy to be low. However, since we use the same classifier across all of the models and steerability approaches, we believe that the metric is fair i.e. the model ranking is reliable. The steerability test and metric that we measured is designed to be a difficult test that aligns with what people typically seek in practice. The reason we focused on quantitative results rather than visualization is because we found that previous work on steerability often showed only cherry-picked examples; we believe that quantitative results are better for ranking different methods.
> >
> > **Re: Questions**
> > ___
> >
> > - **Scaling the number of concepts**:  Please find our response in the weakness section. We did not find any performance degradation in the concept-bottleneck generative model as we scaled the model.
> > - **Are concepts equal**: In the training phase, we did not weight concepts differently (we didn't penalize one concept more than another i.e. $\alpha$ in the loss function equation 4 was not adjusted for each concept separately), so all concepts are equal; however, this is a design choice. During inference/test-time, when an output is generated, the weight of each concept is determined by the output of the probability vector, so some concepts affect the model more than others. In Figure 4, we show a bar chart that indicates how much each concept affects the output of the generative model. For example, in the CB-DDPM trained on Color-mnist, we see that for the generated red digit, it is indeed the red color concept that has the most effect on the model output followed by the digit concept as we would expect. These concept rankings can be used to interpret the output of the model.
> > - **Role of unknown concepts**: The pre-defined concepts alone can be incomplete, so the goal of the unknown embedding is to capture and allow for additional concepts that are necessary for generation but for which are unspecified and that we do not have labels for. For example, in CelebA, we do not control the background that is encoded in image, so it is part of the unknown embedding. In the proposed formulation, we do not allow for intervening on the unknown embeddings, so we can’t steer towards its presence.
> >
> > Thank you for the feedback. We would happy to address any other concerns that you have.

---

> ### Author Response · Authors · 2023-11-21
> **End of discussion approaching**
>
> Dear Reviewer  YbNz,
>
> Given that the end of the discussion period is approaching, we would like to ask if you have any further concerns or questions, particularly as a follow-up to our response?
>
> Thank you in advance!

---

> > ### Comment · Reviewer_YbNz · 2023-11-22
> > **Thank you for your detailed response!**
> >
> > I appreciate the time taken to address the concerns I and the other reviewers raised. I found the responses mostly satisfactory and I am raising my score to 6 - I still think this work can be further refined and improved, but I do think there is an interesting contribution here which the ICLR community can benefit from.

---

> > > ### Author Response · Authors · 2023-11-22
> > > **Thank you**
> > >
> > > Thank you for the follow-up and updated score! We appreciate the discussion, your feedback, and your suggestions.

---

### Official Review · Reviewer_9WtF · 2023-11-01

**Soundness:** 3 good
**Presentation:** 3 good
**Contribution:** 3 good
**Rating:** 6
**Confidence:** 3

**Summary:**

This paper studied concept bottleneck on a number of generative models, like VAE, GAN, and Diffusion. The work shows using concept bottleneck enables steering generative models and debugging better.

**Strengths:**

1. The idea of introducing a concept bottleneck layer to generative models is very interesting. While this has been studied in discriminative models, this is the first time that the reviewer see concept bottleneck been used in generation, which provides explanation and control over the model generation.
2. It shows better steerability than existing methods, like InfoGAN. Based on Table 1, CB-GAN has higher accuracy of the concept classifier. InfoGAN was a distangled GAN that allow control over concept in an unsupervised way.
3. Experiments are conducted on three types of generative models, which shows the method is general.

**Weaknesses:**

Lacking baseline comparison with SOTA model. Without concept bottleneck, exiting work can also control concept, like text-image diffusion, GANSpace, StyleGAN, and others.

Like one can find a direction in StyleGAN that control specified concept, which is also very effective.

Similarly, show comparison to stable diffusion, GANSpace.

**Questions:**

1. How does the method tackle unseen concepts? Can this be extended to open-world concepts?

2. How long does diffusion training take? Do you train all generative models from scratch?

---

> ### Author Response · Authors · 2023-11-18
> **Response to Reviewer 9WtF**
>
> Thank you for the detailed reading and feedback. First, we address the weaknesses and then respond to your questions point-by-point.
>
> **Re: Weaknesses**
> ___
> - **Comparison to SOTA baselines**: As we stated in the general comment, our goal is to use an ***intrinsically interpretable component of a generative model to both steer, interpret, and debug the model***. As it stands, none of the baselines (stable diffusion, GANSpace, etc) accomplishes these goals. We now discuss each baseline in turn.
>
>     - **Text-to-image models (e.g. stable diffusion)**: These models allow for fine-grained control with text inputs. However, given a trained text-to-image model, it is still unclear what features the model relies on to generate its output. While these models allow for text-based control, they do not provide any insight into the models; hence, they are difficult to debug. As an example, in recent work, Tong and Jones et. al. (2023) show that text-to-image models can easily ignore critical words in an input prompt. For example, a Stable Diffusion XL model asked to generate: “a woman proposing to a man” instead produces an image of a man proposing to a woman. As we show in Figure 18, a concept bottleneck model inserted in a diffusion model can help identify the features the model bases its generation on, and hence can immediately point to how to address deficiencies with the model.
>
>     - **GANSpace**: As we stated, our goal is not to do post-hoc interpretation but to insert an intrinsically interpretable layer into the generative model. GANSpace uses PCA to identify the key directions/features in the representations of a GAN. The approach can then vary the representations along these directions to vary the output of the model. Our approach differs from theirs in several ways: 1) since we insert an interpretable layer into the model, we do not need to search for the representations in order to control them. 2) It is possible that the model representations might not encode for a feature that one might want to control. For example, in the Celeb-A setting, we might be interested in controlling the presence of Glasses. However, it is possible that none of the most important PCA directions correspond to Glasses. In contrast, our proposal directly encodes the `glasses' concept as part of the model’s latent representation, so we can immediately identify whether the model has been able to learn that concept.
>
>     - **StyleGAN:** We agree that the StyleGAN model has been used to demonstrate impressive steering/control capabilities. Controlling styleGAN typically can be done in different ways, either by conditioning where the concepts are concatenated to the latent vector, classifier guidance, or model editing methods such as GANSpace as discussed previously. Such methods enable steerability (although we found that for steerability CB layer outperforms conditioning and classifier guided methods as shown in Table 1), but it does not allow for interpretability (i.e we do not understand which concepts the model relied on during generation) and debuggability (i.e we can not tell if the model actually learned a certain concept during training). An alternative approach is adding a CB layer to styleGAN, by doing this, we can explicitly learn the desired concept, we eliminate the need to search for weights correlated with concepts as we know ahead of time which parameters will encode for a particular concept, the concept layer allows us to debug and interpret the generative model during and after training. During training, we can identify which concepts the model is currently unable to encode, and post-training, we can use the output of the concept layer to identify the key features responsible for the model’s output. To address your concern, we added a section in the appendix explaining how one can create CB-StyleGAN; please find the full experiment and results in Appendix B.1.1 (in the updated paper).
>
> **Re: Questions**
> ___
> - **Unseen Concepts**: As we discuss in the paper, we partition concepts into two categories: known (pre-determined human understandable ones), and unknown concepts. The unknown concepts indeed directly encode for unseen concepts for which the model designer did not specify ahead of time in the known set. Your suggestion about open world concepts is interesting, we did not consider it in our initial formulation. However, we can easily adapt the current formulation to allow for a dynamic set of concepts. We leave an extension to open-world concepts for future work.
>
> - **Diffusion Model**: Training diffusion on LAION required 240 V100 GPU hours, training diffusion on Celeb-A required 112 V100 GPU hours. We have updated the manuscript to include the compute time.
>
> - **Training from scratch:** Yes all models are trained from scratch.
>
>
> **References**
> - Shengbang Tong, Erik Jones, and Jacob Steinhardt. Mass-producing failures of multimodal systems with
> language models. arXiv preprint arXiv:2306.12105, 2023.

---

> ### Author Response · Authors · 2023-11-21
> **End of discussion approaching**
>
> Dear Reviewer  9WtF,
>
> Given that the end of the discussion period is approaching, we would like to ask if you have any further concerns or questions, particularly as a follow-up to our response?
>
> Thank you in advance!

---

### Official Review · Reviewer_vC1Y · 2023-11-02

**Soundness:** 3 good
**Presentation:** 3 good
**Contribution:** 3 good
**Rating:** 6
**Confidence:** 3

**Summary:**

The paper proposes a concept bottleneck (CB) based approach to make generative models interpretable. Specifically, it proposes to insert a CB layer into generative models to constraint (most of) their internal representation to a certain concept set, making it quantifiable with respect to the concepts. The key challenge the paper highlight is that features to perform generative modeling may not be always interpretable. To address this, the paper propose to extend previous Concept Embedding Models (CEMs) to additionally encode an "unknown" concept that is meant to be orthogonal to any other concepts. The experimental results show that the proposed approach achieves better accuracy on steering concepts compared to other conditional generative modeling approaches, and providing additional features such as interpretability and debugging of the individual models.

**Strengths:**

- The paper addresses an important yet under-explored problem of interpreting generative models.
- I found the idea of introducing unknown concept vector is interesting and novel.
- The proposed method is widely applicable for diverse model families, including VAEs, GANs, and Diffusion models.
- The paper also partially demonstrates the scalability of the method with a LAION subset on Diffusion model, following contemporary practices.
- The effectiveness of the method is clearly validated through experiments.

**Weaknesses:**

- The paper claims at Abstract and Introduction that the proposed method is model agnostic, but I am not certain on that. For example, it seems to me that applying the CB layer to other types of generative models can be non-trivial, e.g., to normalizing flows and invertible models.
- Although the paper presents some scalability experiments, the other parts of the experiments can be seen as somewhat limited in their scale. For example, the paper only explores simple GAN architectures not covering modern ones such as StyleGAN-2. I am also wondering why more recent methods such as GAN inversion could not be a baseline. If these point can be addressed, it will be also beneficial to support the model-agnostic aspect of the method.
- More qualitative comparisons could be added and highlighted for the steerability experiments, given that the quantitative results are dependent on the pre-trained concept classifiers.
- (minor) The definition of concept orthogonality loss, Eq (5), is written in somewhat confusing manner - e.g., the index j is not used in the definition.

**Questions:**

- The paper mentions Platt and Barr (1987) regarding hyper-parameter optimization. Does it mean that the paper actually applied that method for tuning in the experiments? If so, it may be good to provide an overview on the method in the paper as well.
- It seems making sure the minimal concept orthogonality loss is crucial for the soundness of the method, otherwise there can be an overlap between the given concepts and the "unknown" concept. One immediate ablation one can try is to check whether the loss is minimized is to strictly project the learned unknown context vector to be orthogonal to other vectors and see if there is degradation in performance. Or I think this kind of procedure can be even incorporated into the training phase to guarantee the minimal concept orthogonality loss. Any discussion regarding the actual orthogonality of the learned unknown context vector would be helpful for the readers.

---

> ### Author Response · Authors · 2023-11-18
> **Response to Reviewer vC1Y**
>
> Thank you for the detailed reading and feedback. First, we address the weaknesses and then respond to your questions point-by-point.
>
> **Re: Weaknesses**
> ___
>
> - **Abstract, Introduction, and Normalization Flows**: Thank you for raising this important point. Indeed, we tested three types of generative models in the draft. However, we found your suggestion of normalizing flows and invertible models intriguing, so we tested our proposed approach on a Normalizing flow model trained on Color MNIST, we added this experiment in Section B.1.2 of the Appendix. Our CB layer proposal translates, as is, to such models as well. We show examples generated from the model in Section B.1.2 of the Appendix. We expect future work to test our findings more exhaustively and extend them to other families of generative models. We take your feedback to mean that we should explicitly specify the class of models we focus on, which we have done in the updated manuscript.
>
> - **Not covering modern ones such as StyleGAN2:** As per your request, we added a CB layer to StyleGAN2 in Appendix B.1.1 (in the updated paper). For StyleGAN, the CB layer was added after the mapping network in the generator, an encoder was also added before the generator to ensure that the generated image matches the concepts constrained in the loss, the exact architecture is shown in appendix B.1.1 Figure 7 (a). We trained StyleGAN2 on Celeb-A 64x64 and CUB 128x128. We added images generated by CB-styleGAN2 for both datasets in the appendix Figure 7 (b) and reported FIDs as well. We find that using more complicated architectures improves the quality of the generated image (as expected) and inserting a CB layer does not degrade the generation quality.
>
> - **GAN inversion baseline**: As we mentioned in the general comment, while GAN inversion approaches enable a model to be steered, these approaches do not address the goals that we set forth to tackle in this work. To re-iterate: **we seek an intrinsically interpretable component of a generative model that allows us to steer and interpret the output of the generative model simultaneously.** Meanwhile, GAN inversion approaches do not provide any form of understanding or insight into which features the model bases its generation on. In addition, we require that the approach be intrinsic to the model and not a post-hoc approach. Critically, a GAN inversion approach cannot indicate the key features that a model relies on for its output. Similarly, to the best of our knowledge, most GAN inversion techniques do not explicitly insert an interpretable component into the model. We think the GAN inversion approaches address an important challenge; however, our goals are orthogonal to these challenges.
>
> - **Qualitative Results:**  As you requested, we updated the Appendix to include several additional qualitative steerability experiments from StyleGAN2; see  Figure 11 in the appendix.
>
> - **Concept Orthogonality Loss:** Thank you for noting this issue, $j$ is the sample in the mini-batch, we have updated the main text under Eq 5 to clarify this.
>
> **Re: Questions**
> ___
>
> - **Optimization**: Thank you for raising this important point. On the VAE model we compared Platt and Barr to a standard grid search/sweep over a hyper-parameter set. We didn’t observe a substantial improvement in model FID scores for the subset of models trained. We used an open-source Pytorch implementation of the Platt and Barr approach, which does not require any modification for the setting considered here. The approach minimizes a loss function that is subject to bound constraints with arbitrary parameters that weight the importance of these functions. We added this discussion in Section B.2 in the appendix; we also referred to the open-source implementation used.
>
> - **Orthogonality:** We agree with your assessment that it is crucial for the unknown concept embedding to be orthogonal to known concept embeddings. Removing orthogonality between the known and unknown concepts causes steerability degradation, as shown in the ablation Table 3. Our current algorithm encourages orthogonality by adding the loss in equation 5 but does not enforce it. We tried forcing orthogonality during training (as you suggested) by projecting the concepts embedding to be orthogonal to non-concepts embedding during training using the Gram-Schmidt Process [1]; we found that this makes training unstable and harms generation. We note that there are other schemes that can achieve independence, including KL minimization, mutual information minimization, etc. We leave exploring different orthogonality schemes as future work.
>
> [1] Schmidt E.Zur Theorie der linearen und nichtlinearen Integralgleichungen I. Teil: Entwicklung willkürlicher Funktionen nach Systemen vorgeschriebener. Mathematische Annalen 1907; 63: 433–476.

---

> ### Author Response · Authors · 2023-11-21
> **End of discussion approaching**
>
> Dear Reviewer vC1Y,
>
> Given that the end of the discussion period is approaching, we would like to ask if you have any further concerns or questions, particularly as a follow-up to our response?
>
> Thank you in advance!

---

### Author Response · Authors · 2023-11-18
**General response part 1**

We thank all the reviewers for their feedback, and for noting that the paper addresses an *important problem* (**Reviewer vC1Y**),  is *interesting* (**Reviewers vC1Y, 9WtF**) and *novel* (**Reviewers 5NVe, vC1Y**), *well-written* (**Reviewers YbNz, 5NVe**), and presents a *general solution* (**Reviewers vC1Y, 9WtF**). Here we address general concerns.

**Our Goal**
___
In this work, we develop a **generative model with an intrinsically interpretable component that can be used to simultaneously: 1) interpret/debug, and 2) steer the output of the model**. It is important that the component be intrinsic to the model, and also enable the two aforementioned capabilities. **To the best of our knowledge, no current approach enables these capabilities**. Below we state a few key advantages of our approach:

- **why did my model generate this output (interpretability)?** In current text-to-image models there is no way to interpret a model’s output, and to figure out which features the model is reliant on. In CBGMs, we can use the output of the concept layer to identify the key features responsible for the model’s output.
- **model debugging**: The CB layer allows us to diagnose whether a concept has been learned. When a text-to-image model is being trained, it is difficult to know which kinds of text prompts the model will err on. However, in our case, one can inspect the loss of the CB layer to know which concepts the model cannot control. To understand whether a text-to-image model is ineffective at controlling a concept, one must generate and inspect several input-output pairs. During training, the concept layer allows us to identify which concepts the model is currently unable to encode, and post-training we can use the output of the concept layer to identify the key features responsible for the model’s output. Current generative models do not provide this capability (See Appendix C for an Example).
- **removes the need for searching for weights (or representations) that correlate with concepts**: another crucial capability that the concept layer provides is that we can localize both the representations and weights responsible for generating a particular concept without searching for the concept.
- **learning explicit concept representations rather than correlational representations**: text-to-image models learn concepts from correlations between images and text, which can often lead to learning spurious associations. The use of the CB layer enables explicit learning of concepts by making the output dependent on the concept embedding .

**Orthogonal Approaches**
___
Several reviewers asked us to contrast our proposal with text-to-image models, GAN inversion, and GANSpace. We now take each approach in turn and discuss why they are not relevant baselines for the problem we tackle.

- **Text-to-image models (stable diffusion)**: despite impressive conditional generation abilities, current text-to-image models donot have intrinsically interpretable components. This means that when these models make mistakes, it is not clear which features they are relying on for the output. Hence, correcting such mistakes is challenging (See Appendix C for a demonstration).  In recent work, Tong and Jones et. al. (2023) show that text-to-image models can easily ignore critical words in an input prompt. For example, a Stable Diffusion XL model asked to generate: “a woman proposing to a man” instead produces an image of a man proposing to a woman. A concept bottleneck model inserted in a diffusion model can help identify the features the model bases its generation on, and hence can immediately point out how to fix deficiencies with the model (See Appendix C).
- **GAN Inversion**: GAN inversion methods allow the reconstruction of the latent code that led to an output. These approaches do not provide insight into which features the model bases its generation on.
- **GANSpace**: uses PCA to identify the key directions in a GAN's representations. It then varies the representations along these directions to modulate the presence of a concept in the output. Our approach differs in several ways: 1) since we insert an interpretable layer into the model, we do not need to search for directions in representations in order to control them. 2) It is possible that the model representations might not encode for a feature that one might want to control. For example, in the Celeb-A setting, we might be interested in controlling the presence of Glasses. However, none of the most important PCA directions might correspond to Glasses. In contrast, our proposal directly encodes the `glasses' concept as part of the model’s latent representation, so we can immediately identify whether the model has been able to learn that concept. Critically, GANSpace does not indicate the key features the model relies on.

Taken together, the above baselines address a different set of challenges than the goal we set out to solve.

---

> ### Author Response · Authors · 2023-11-18
> **General response part 2**
>
> **Other Concerns**
> ___
>  Below we briefly discuss other cross-cutting concerns.
>
> - **Adapting the CB layer to StyleGAN (Reviewer vC1Y and 9WtF):** We added section B.1.1 in the appendix showing how to add a CB layer to StyleGAN2.
> - **Ground-truth Concept Annotations (Reviewer YbNZ and 5NVe)**:  We performed a new set of experiments where we varied the percentage of concept annotation available for CBGMs we added this experiment in Appendix B.3.3. We find that with about 20% of annotations, we can directly match steerability metrics of a fully annotated training set.
> - **Steerability metric** **(Reviewer YbNZ and 5NVe):**  To clarify, we calculate the steerability metric as follows: We start with a high-accuracy classifier trained on real data; we ensure that the minimum accuracy of any concept classifier is at least 98 % on a held-out test set from the real data. We sample noise latent vectors and then use it to generate images from the model; for each concept we pass the generated image to the concept classifier; we save the latent vector if the classifier predicts the concept to be absent (i.e., the probability of the concept present is less than 0.5) we continue this process until we have 1K samples. For each intervening approach (including CB layer), we intervene on a concept and generate new images. We then pass the newly generated images to the classifier again and measure the fraction of inputs for which the indicates greater than 0.5 probability of concept presence, which we report in Table 1.
>
> For a detailed discussion, please refer to the individual point-by-point responses. Note that in the revised manuscript, changes are shown in orange. We look forward to hearing from you and will happily address any remaining feedback!

---

### Meta-Review · Area_Chair_JRhH · 2023-12-05

**Metareview:**

This paper contributes the “concept bottleneck” later for generative models, which allows for better control and introspection of the generative process. The approach follows a similar recipe to Concept Embedding Models (CEMs) with the addition of an “unknown” concept that is meant to be orthogonal to the other concepts (for which labels are available during training). The merits of adding the concept bottleneck is shown for several generative models including GANs, VAEs, and Diffusion models.

The paper was initially received with some critique, especially regarding the motivation of this kind of work, the need to rely on supervised annotations, and the limited significance of the contribution. Several of these concerns were sufficiently addressed during the rebuttal as is evident from multiple reviewers raising their score from “marginally below” to “marginally above”. The authors also did a good job at discussing possible use cases for this kind of work, and arguing for the overall benefit to using such an approach as opposed to using text-conditioning.
On the other hand, the originality and significance of the contribution remains somewhat limited.

Given broad agreement to accept among the reviewer (albeit marginally), and taking into account the strengths of the author response (addressing nearly all concerns) and the included changes that have clearly improved the paper, I am recommending to accept too. For a future revision, I encourage the authors to revisit the paper once more and include as many of their comments regarding motivation and utility in the paper as possible

**Justification For Why Not Higher Score:**

This paper is borderline as it is, and the somewhat limited originality and significance (as pointed out by the reviewers) prevent me from recommending a higher score.

**Justification For Why Not Lower Score:**

Multiple reviewers engaged with the authors during the author-reviewer discussion and were convinced by the authors to increase their score. Having read the author response myself, I agree it clarifies a lot and addresses many of the issues that were outlined (as far as they can be addressed). Overall I agree with the reviewer assessment that this paper contributes and interesting analysis of applying CEM-like techniques to generative models and makes a good case for the benefit of this approach (relative to VLMs), despite requiring annotated data.

---

### Decision · Program_Chairs · 2024-01-16

Accept (poster)